# Micro- and Nanostructured Fibrous Composites via Electro-Fluid Dynamics: Design and Applications for Brain

**DOI:** 10.3390/pharmaceutics16010134

**Published:** 2024-01-19

**Authors:** Nergis Zeynep Renkler, Stefania Scialla, Teresa Russo, Ugo D’Amora, Iriczalli Cruz-Maya, Roberto De Santis, Vincenzo Guarino

**Affiliations:** 1Institute of Polymers, Composites and Biomaterials (IPCB), National Research Council of Italy, Mostra d’Oltremare Pad. 20, Viale J.F. Kennedy 54, 80125 Naples, Italystefania.scialla@cnr.it (S.S.); cdiriczalli@gmail.com (I.C.-M.);; 2Department of Chemical, Materials and Industrial Production Engineering, University of Naples Federico II, 80125 Naples, Italy

**Keywords:** electro-fluid dynamics, nanofibers, nanoparticles, fibrogels, brain

## Abstract

The brain consists of an interconnected network of neurons tightly packed in the extracellular matrix (ECM) to form complex and heterogeneous composite tissue. According to recent biomimicry approaches that consider biological features as active components of biomaterials, designing a highly reproducible microenvironment for brain cells can represent a key tool for tissue repair and regeneration. Indeed, this is crucial to support cell growth, mitigate inflammation phenomena and provide adequate structural properties needed to support the damaged tissue, corroborating the activity of the vascular network and ultimately the functionality of neurons. In this context, electro-fluid dynamic techniques (EFDTs), i.e., electrospinning, electrospraying and related techniques, offer the opportunity to engineer a wide variety of composite substrates by integrating fibers, particles, and hydrogels at different scales—from several hundred microns down to tens of nanometers—for the generation of countless patterns of physical and biochemical cues suitable for influencing the in vitro response of coexistent brain cell populations mediated by the surrounding microenvironment. In this review, an overview of the different technological approaches—based on EFDTs—for engineering fibrous and/or particle-loaded composite substrates will be proposed. The second section of this review will primarily focus on describing current and future approaches to the use of composites for brain applications, ranging from therapeutic to diagnostic/theranostic use and from repair to regeneration, with the ultimate goal of providing insightful information to guide future research efforts toward the development of more efficient and reliable solutions.

## 1. Introduction

Tissue damage may result from different brain disorders or issues, including stroke, brain tumors, traumatic brain injury (TBI), and neurodegenerative diseases [1]. Depending on the reason and severity of the injury or disease, there are many ways to deal with treating brain tissue damage, including surgery and medication. Even though early detection is essential for controlling the symptoms of many brain diseases, there are still difficulties and restrictions in treating brain tissue damage. Currently, there are no treatments that can stop the progression of neurodegeneration in the brain or completely restore the function of damaged brain tissue, which is a major problem facing neuroscience and neurology [2]. First, the blood–brain barrier (BBB) may restrict the delivery of therapeutic medicines to areas of brain injury [3], and brain tissue has limited self-healing and regeneration ability [4]. Furthermore, the complex nature of the brain and the different characteristics of various lesions make it difficult to design efficient and focused therapies for brain tissue injuries [5,6]. There is a need to improve treatment efficiency for brain tissue damage; in fact, many studies are addressing these challenges and continue to explore innovative approaches to this issue by comprehensively understanding the pathophysiology of brain injuries and developing alternative or adjunctive therapies to improve outcomes [7,8]. For instance, human pluripotent stem cells (hPSCs) draw attention as neuronal replacement strategies, and success has been achieved from exogenous and endogenous cell sources [9,10]. Additionally, the capacity to generate new neurons from non-neuronal cell sources opens up new avenues for the development of regeneration-based treatments for degenerative illnesses and traumatic brain and spinal cord injuries [11]. However, poor cell survival, functional integration of transplanted cells into the host tissue, the need to overcome multiple barriers in delivering therapeutic cells to the brain, and recapitulating the complex architecture of the brain remain limited [12,13]. To overcome these challenges, strategies to enhance brain regeneration by promoting neurite and axonal growth or stem cell differentiation and migration to the appropriate site are required. From this perspective, tissue engineering holds promise due to its feasibility in addressing brain tissue damage by creating an artificial environment for neural cell growth and differentiation.

Research on functional bioengineered models will likely focus on developing in vitro models or systems that can represent the function and structure of the brain and allow in vitro assessment of neural development and pathology of damaged tissue. Microfluidic devices, organoids, spheroids, and scaffold-based or bioprinted constructs that integrate materials and cells can be used to mimic parts of the CNS [14]. To create a microenvironment for neural cells, it is crucial to understand the structure and function of the extracellular matrix (ECM). The ECM maintains structural integrity through a dynamic, three-dimensional (3D) network of macromolecules, facilitates cell communication, and plays a vital role in various cellular functions [15]. Furthermore, the ECM directs tissue development and regeneration by providing spatial and temporal control of biological, physical, and chemotactic cues [16]. The natural ECM of the brain is unique and specialized to support the complex structure and function of neural tissue.

From this perspective, bioinspired materials can be selectively engineered for the brain and intended to interact with specific biological systems, including brain cells. Novel biomaterials are currently being investigated for their potential to enhance neuronal regeneration, inflammatory regulation, and synaptic plasticity in the post-stroke brain, as well as for the treatment of neuropathies. Significant attention is being paid towards the development of scaffolds capable of replicating key functionalities of the brain ECM [17,18]. For instance, the fabrication of hybrid materials combining both natural and synthetic components is growing, which allows leveraging the advantages of each phase to more closely mimic the composition of the brain matter [19,20,21]. Alternatively, different polymers were recently functionalized to better imitate the mechanical and biochemical attributes of the brain ECM [22,23]. Among them, electroconductive polymers are particularly useful for CNS, since they are electrically sensitive tissues, whereas hydrogel-based biopolymers with ionotropic behavior show promise in neuroregeneration in combination with stem/precursor cells [24,25].

However, the fabrication of bioactive scaffolds that can provide a selected pattern of signals (e.g., topological, chemical, physical) to brain cells remains a great challenge for medicine, especially from the perspective of clinical transition. Once implanted into the brain tissue, they should support regeneration, adapt to the injury geometry, and be capable of providing cell migration cues [26]. Moreover, due to the integration of bioactive molecules/factors to be delivered locally, they should further contribute to the definition of therapeutic treatments suitable for directly reprogramming secondary cells of brains, such as astrocytes, into neurons [27].

In this context, electro-fluid dynamic techniques (EFDTs), such as electrospinning (ES), electrospraying, and other related methodologies, present a valuable opportunity for the precise engineering of a diverse range of composite substrates aimed at emulating the intricate structure and composition of the brain ECM [28,29]. For instance, electrospinning allows the creation of nanofibrous structures by electrostatic interaction with polymer melts and/or solutions that closely mimic the fibrous architecture of naturally occurring extracellular matrix [30]. Similarly, electrospraying—i.e., atomization of liquid solutions into micro/nanosized droplets via the application of a high-voltage electric field [31]—may offer an easier route to incorporate certain biochemical cues into the fibrous network, thus improving bioactive functionalities needed for mimicking the complex molecular pattern of the brain [31]. Furthermore, the high versatility of EFDTs may also allow the designing of a large set of highly customizable devices to be used as frameworks for the methodical development and production of biomimetic materials with highly modulable characteristics toward a personalized approach for brain repair/regeneration.

In order to investigate neurological diseases, test possible drugs, and develop novel treatments, even more realistic and effective models of the brain ECM must be created. Here, we discuss current developments in the production of biomaterials using EFDTs, which may result in methods for brain regeneration and repair. We first introduce the fundamentals of EFDTs and highlight their practical applications to brain tissue for appropriate purposes of polymer micro/nanocomposite materials. Finally, we outline the current challenges and future perspectives.

## 2. Electro-Fluid Dynamic Techniques

EFDTs involve the application of electrical forces to manipulate and control the behavior of polymeric materials in solution to achieve specific properties and structures. They have been utilized to process/synthesize various polymer materials, including porous thin films, nanofibers, nanorods, ribbons, particles, and capsules of different sizes and shapes (Figure 1) [32]. Among them, electrospinning has been extensively employed in the biomedical field due to its unique ability to produce nanofibers with tailored morphological properties at the micro- and sub-micrometric dimensional scale [33]. Moreover, electrospraying has been widely studied for the fabrication of drug-loaded micro- and nanoparticles with different architecture (e.g., mono/multiple phase, Janus, core–shell, etc.) for pharmaceutical use, offering potential advancements in drug delivery systems (DDSs) [31,34].

The high versatility of EFDTs processing modes fits well with the use of biomaterials with highly tailored chemistry, thus providing a fine matching between structural and functional properties. This allows for the development of advanced composite devices for applications such as repair, regeneration, drug delivery, biosensing and diagnostic/theranostic purposes.

### 2.1. Electrospinning

Electrospinning is a technique that transforms polymer solutions into micro- or sub-micro dimensioned fibers under the effect of a high-voltage electric field, making it as a versatile method for fabricating nanofiber materials comprised of various polymers and ceramics [35]. Electrospinning is known for its simplicity, flexibility, and multipurpose nature in fabricating submicron-scale fibers [36]. The basic setup of electrospinning comprises the key components of a high-voltage power supply, a needle spinneret, and a grounded collector, alongside key parameters contributing to the successful fabrication of nanofibers (Figure 2) [28].

The process involves employing a polymer solution or melt, which should have the appropriate viscosity and conductivity for effective electrospinning. The droplet of polymer solution at the needle tip, subjected to a high-voltage electric field sourced from the power supply, transforms into a cone shape (known as a Taylor cone). By creating an electric field between the syringe tip (spinneret) and a grounded collector, the Taylor cone deforms for the creation of a charged jet. This charged jet subsequently stretches and thins due to the electrostatic forces. As the jet moves toward the grounded collector, solvent evaporation occurs, leading to polymer solidification and accumulation on a grounded or oppositely charged collector. This process results in the formation of nanofibers, presenting as a nonwoven mat or mesh [37,38]. Fiber morphology, structures and functions can be significantly influenced by the process, solution, and the environmental parameters [39].

One of the advantages of this method is its ability to combine a variety of natural and synthetic polymers, along with inorganic materials, to effectively produce micro/nanocomposite fibers with the multifunctionality that they bring. Moreover, the high-surface area, porosity, and the capability of loading drugs or other biomolecules into the fibers make them useful for different applications [40,41]. In tissue engineering and biomedical applications, the fabrication of electrospun nanofibers using both synthetic and natural polymers has gained significant attention. The use of synthetic polymers in electrospinning is advantageous due to their mechanical stability, biocompatibility, biodegradability, and non-toxicity, making them suitable for biomedical applications. However, many synthetic polymers usually processed by electrospinning often require the use of chemically aggressive organic solvents, with relevant limitations in the manufacturing of cell-friendly substrates. Accordingly, the main constraints associated with the use of toxic solvents in electrospinning are paving the way to the use of less or non-toxic solvents (e.g., water, ionic liquids), so limiting the use to a restricted group of green polymers with more sustainable properties (e.g., PVA, PEO). Alternatively, they are increasingly used in combination with natural polymers. such as proteins of polysaccharides with native function biological recognition [42,43].

Additionally, by a large customization of the electrospinning setup (i.e., needles, collectors) (Figure 2), it is possible to fabricate scaffolds with a controlled spatial distribution of fibers, in order to meet the structural organization of different tissue types (e.g., muscle, bone, skin, nerve).

Based on this, electrospun fibers have gained significant attention for brain tissue applications, particularly for their morphological similarities with the structure of the neural tissue ECM at the nanoscale level and their high porosity. This provided a contact guidance for regenerating axons and can be engineered to generate a desired glial cell response to damage in the brain by adjusting a variety of properties, including fiber alignment, diameter, surface nanotopography, and surface chemistry [44]. The influence of various parameters, such as solution properties and electrospinning process parameters (Figure 3), on fiber formation and structure has been widely investigated.

It is known that fiber diameter and surface topography sufficiently affect neural stem cell differentiation and proliferation. Higher levels of proliferation and cell spreading of neural stem/progenitor cells have been observed when the fiber diameter decreased [47]. Moreover, the ability to orient neurite growth in electrospun fibrous neural conduits demonstrates the impact of fiber diameter on directional growth, emphasizing the significance of tailored fibers in guiding tissue regeneration [48]. Also, Mahairaki et al. showed that fiber alignment has an effect on the behavior of human neural progenitors. On aligned fibers, polarized cell morphology extends along the axis of the fiber, whereas on random fiber substrates, they form non-polarized neurite networks [49]. Another study has shown that mouse embryonic stem cell morphology on randomly oriented fibers caused short-range topographical guidance, whilst cells on oriented fibers exhibited long extension and neuron outgrowth along the fiber, since oriented fibers provided more contact guidance [50,51]. In this context, the use of different setup configurations and process modes (e.g., triaxial, quad-fluid coaxial, tri-fluid side by side, and coaxial electrospinning) may efficiently enable the fabrication of advanced functional materials with more complex architecture and material phase organization, with peculiar features for mimicking the brain microenvironment [52]. Alternative approaches also suggested the use of polymer films to be used for partially masking the surrounding collector, thus providing a confinement of fibers into selected portions/regions with peculiar shape and geometries in order to create controlled patterned areas [53].

### 2.2. Electrospraying

Neurodegenerative diseases, traumatic injuries, and stroke cause damage to the human nervous system. For instance, the increased prevalence of neurodegenerative diseases and low efficiency of treatments due to the presence of the BBB, limiting the penetration of drugs toward the CNS, has led researchers to explore new strategies for the development of biocompatible devices able to locally deliver the active molecule while decreasing the characteristic side effects when systemically administrated [54,55].

Electrospraying is an EFDT that involves the liquid atomization of a polymeric solution influenced by a high voltage, as in the case of the electrospinning technique. The electrospraying technique has been used for the fabrication of micro- or nanoparticles employed in several biomedical applications, particularly for DDSs [56,57,58]. The main difference between both EFDTs is related to the polymer concentration (Figure 3). For electrospraying, low concentrations are preferred to allow the jet to break into small liquid droplets. The formed droplets are highly and identically charged, which prevents their agglomeration and allows their dispersion in the space [59].

The basic setup is similar to the electrospinning technology and consists of a high-voltage power supply, a syringe with a metallic needle connected to a syringe pump and the grounded collector (Figure 2). During the electrospraying process, electrical forces overcome the surface tension of the droplet under the influence of an external electric field. Moreover, solvent evaporation allows the collection of solid particles. The size, morphology, and surface of particles can be modified and controlled by process parameters such as flow rate, applied voltage, and polymer concentration [31,60]. For instance, different morphologies of particles, such as spheres, donut-like, and corrugated shapes with sizes ranging from several tens of microns to hundreds of nanometers, were observed to be influenced by polymer molecular weight and concentration, solution flow rate, voltage, and solvent [61].

In particular, micro- or nanoparticles via electrospraying have been fabricated for drug delivery applications, due to their high loading efficiency and narrow size distribution. Electrospraying is a one-step process, compatible with different biomaterials (e.g., natural and synthetic polymers) that can act as carriers of a wide variety of molecules and drugs (e.g., water or poorly water-soluble drugs) [62,63]. Congruently with electrospinning, the use of organic solvents in electrospraying poses some limitations, as it can potentially harm the bioactivity of biomolecules (i.e., proteins, genes, enzymes), also compromising the transmembrane interaction with cells. However, the electrospray is a process mainly driven by the solvent evaporation, so that the use of harmless or aqueous solvents impose to adopt further solutions, such as the use of additives or other compounds, to promote efficient interactions of polymer solution with electrical forces in the face of some effects in terms of particle size—only micrometric and shape not spherical/eccentric [64]. For the particular case of brain or neural injuries, drug-loaded particles have been proposed to implant and locally release the therapeutic agents; therefore, the drug can bypass the BBB and increase the concentration of the molecule in the brain [65]. For instance, paclitaxel-loaded biodegradable particles with controllable size and morphology were fabricated by electrospraying with high encapsulation efficiency for local drug delivery in vitro more than 30 days in order to treat malignant glioma [61]. The high versatility of electrospraying has allowed its use in combination with other methodologies. For instance, paclitaxel-loaded poly(lactic-co-glycolic acid) (PLGA) microspheres have been entrapped in electrosprayed alginate hydrogels, which allowed drug delivery with near zero-order release and low burst release to treat malignant brain tumors with the critical advantages of implantability and sustained release [66].

Tissue regeneration in diseases such as cancer is characterized as a multistage process; thus, more complex and specific delivery systems have been studied. In this regard, coaxial electrospraying is a good alternative for developing carriers with high encapsulation efficiency and capability to entrap two or more different active molecules to release them sequentially [67]. For instance, two drugs with different hydrophilic properties (e.g., paclitaxel and suramin) were encapsulated in core–shell microspheres at different ratios. The results showed advantages and possible applications for the treatment of brain tumors with tailored properties according to the properties of the active molecules [68].

Beyond mimicking the peculiar architecture of the ECM, it has been demonstrated that the addition of bioactive cues via nanoparticles can promote cell differentiation, so guiding the regeneration process [69]. From this perspective, micro- or nanoparticles have been widely used to support brain regeneration through the use of nanoparticles, including specific chemical (e.g., growth factors [70]) or topological signals [71].

## 3. From Mono- to Multicomponent Substrates

It has been demonstrated that electrospinning is a practical and economic method for mimicking the shape, functional surface properties, and chemical structure of native tissues [72]. Fibers can be tailored according to their diameter, porosity, orientation, layering, surface structure, mechanical qualities, and biodegradability. To meet the properties of complex tissues, cooperative use of several types of polymers is quite promising, since the combination of these polymers can provide new materials with characteristics that are strikingly close to those of natural tissues. In fact, the use of various polymers inside multicomponent scaffolds has become an appealing strategy, providing a flexible and beneficial combination of material qualities appropriate for a range of applications. This customization extends to the use of different materials and functional molecules, resulting in a profusion of biomaterials with diverse properties that are yet to be fully explored [73]. There are various ways to obtain fibers containing more than one component in terms of material and process (Figure 4).

Through the electrospinning technique, two or more distinct materials—typically polymers or polymer solutions—are combined. In addition, it can include additives and nanomaterials to produce fibers that have various desired properties either before or during the electrospinning procedure. Several materials are mixed in a single solution or melt during blend electrospinning. The process is simple: the mixed solution is electrospun to produce composite fibers. Prior to electrospinning, the components must be well mixed and homogenized. Hybrid biomaterials made of both synthetic and naturally generated polymers have been extensively researched to achieve this aim. Especially in tissue engineering, these composites are widely utilized to build scaffolds with desired mechanical and biological characteristics. For instance, because of its biocompatibility, biodegradability, and low immunogenicity, chitosan (CS) is widely employed in biomedical applications with synthetic polymers, like polyvinyl alcohol (PVA), poly(ethylene oxide) (PEO), and PLGA [74]. Also, the blending of polyurethane (PU) and gelatin has been shown to enhance the mechanical, physicochemical, and biological properties of electrospun nanofibers [75]. However, for tissues such as neural tissue, proper conductivity is required to replicate the ECM and regeneration of brain tissue [76]. In fact, blending polymers with conductive materials such as aniline oligomers, carbon nanotubes/fibers, and gold nanoparticles/wires has attracted attention for its potential to overcome the lack of conductive properties of natural/synthetic blend fibers. In addition, as drug carriers, blending electrospun nanofibers is a promising approach for enhancing DDSs. Electrospun multicomponent nanofibers offer a high surface area/volume ratio, tunable porosity, and the ability to encapsulate and release drugs in a controlled manner [74,77]. Complex regenerative or reconstructive processes cannot always be achieved with electrospun fibers by simple drug loading or pure surface alteration.

Blending electrospinning is the process of employing a single needle to electrospun a combination of two or more distinct materials into a single, homogeneous, single-phase working liquid. Multicomponent fibers can be produced by combining two or more materials in the electrospinning process using various variants and procedures. For instance, in coaxial electrospinning using concentrically organized coaxial needles or spinnerets, two or more materials are electrospun concurrently. This enables the construction of core–shell structures, in which the shell is made of a different material and the core is made of a different material. When encapsulating delicate molecules, such as medications or bioactive chemicals, inside protective polymer shells, coaxial electrospinning is highly beneficial [78]. In emulsion electrospinning, one substance is distributed as droplets in another immiscible fluid during the electrospinning process. The emulsion is electrospun to create fibers with a continuous phase encased in a scattered phase. When combining ingredients that might not dissolve or mix well in the same solvent, this approach works well [79]. In addition, electrospinning of two materials concurrently from different spinnerets placed next to each other is known as side-by-side electrospinning. Each substance is in a discrete zone along the length of the fibers produced. This technique works well for making fibers when the interfaces between various components are clearly visible. Side-by-side spinning of silk fibroin and poly-l-lactide (PLLA) has been achieved through electrospinning, demonstrating the versatility of this method in fabricating composite biofibers [80]. The multi-jet electrospinning technique uses many spinnerets, each of which is attached to a different material reservoir. Fibers comprising various materials can be simultaneously deposited on the collector by adjusting the spinneret positions and electrospinning settings [81]. In addition, needleless multi-jet electrospinning has been developed to avoid nozzle clogging and improve productivity by forming a large number of jets under a high electrical field, and multi-material electrospinning can also be accomplished using electrospinning techniques such as electroblowing or electrospraying [82]. During the process, different materials can be blended by changing the configuration or employing numerous nozzles. Illner et al. showed that varying the polymer solution composition and the electrospinning setup, two-component nonwoven structures with tailored properties in terms of flexibility and fiber interconnection can be obtained through a complex manufacturing process [83]. Moreover, Smith et al. reported that combining electrospinning and additive manufacturing (AM) techniques, in particular, multilayered designs with alternating fibers, AM components, and fiber-reinforced bioinks, were created as hybrid scaffolds [84]. The details of the application and the intended qualities of the composite fibers determine which approach is best.

Advanced scaffolds have been developed through the techniques of multi-material electrospinning, which has allowed for the mimicking of the heterogeneity present in native ECMs, increased scaffold porosity for better cell penetration and the synergistic integration of many properties into a scaffold.

## 4. Design of Composite Fibers for Brain

### 4.1. Blended Fibers

Researchers can produce materials with specialized and synergistic qualities by manufacturing nanofibers with multiple components, which makes them ideal for a wide range of applications. In the context of tissue engineering, fibers play a crucial role, and blended electrospun fibers have shown promise for various biomedical applications, including brain tissue engineering. Neural cells have been shown to locate and utilize the cues provided by fibers for migration into hybrid matrices, indicating the potential for tailored fibers to guide cellular behavior [85].

The specific components used rely on several variables, including mechanical qualities, biocompatibility, chemical properties, conductivity, and the intended use of the neural tissue engineering scaffold [86]. By altering the mechanical properties of the final product, blend materials can be created to incorporate a variety of capabilities for uses like controlled drug delivery. It is possible to introduce extra characteristics like conductivity to affect biological activity by modifying fiber chemical features. Moreover, other components enable the delivery of topographic signals to CNS cells, hence broadening the scope of possible uses (Figure 5a). Moreover, multicomponent integration can improve the overall stability and strength of nanofibers and by applying it with different collector assemblies, alignment as well as fiber morphology can be controlled (Figure 5b).

For instance, the fabrication of electrospun scaffolds for neural tissue engineering that blend hydrophilic and hydrophobic materials enable the creation of a balance of features, including enhanced cell contact and structural stability. Lins et al. described the fabrication process and emphasized the possible applications of poly(lactic acid) (PLA)–poly(lactide-b-ethylene glycol-b-lactide) block copolymer (PELA) and PLA–polyethylene glycol (PEG) blends on the morphology, wettability, and mechanical properties of the material and the behavior of neural stem cells (NSCs). The findings suggested that electrospun blend of PLA and PELA have favorable surface characteristics, potentially resembling brain structures [88]. According to another study, composite nanofibers of polyhydroxyphenylvalerate–PCL (PHPV/PCL) improve the mechano-responsiveness and life span of hiPSC-derived cortical neurons [89]. It is possible to create a material with enhanced or balanced performance qualities by blending synthetic and natural components. Proteins and polysaccharides are examples of natural materials that frequently show great biocompatibility, indicating that living cells can tolerate them well. The overall biocompatibility of a material can be improved by blending them with synthetic materials. Saracino et al. examined how the physical characteristics of PCL-based blended fibers affect astrocyte behavior. Blending PCL with a gelatin protein has shown a favorable effect on spreading and alignment of the cell along the fibers, and this sheds light on the way astrocytes interact with the electrospun scaffold and suggests possible uses for PCL–gelatin blended materials with aligned and randomly oriented morphology in regenerative medicine and neural tissue engineering (Figure 5b) [90].

Blended nanofibers with conductive elements, like graphene or carbon nanotubes, can be developed for use in brain tissue or bioelectronic device applications. This gives the nanofiber electrical conductivity, enabling it to be used as an interface with CNS cells. For example, with an emphasis on neurological prosthesis, Bianco et al. investigated the application of single-walled carbon nanotubes (SWNTs) and high-purity carbon nanofibers (CNFs) in electrospun PCL scaffolds. Rat cerebral-microvascular endothelial cells (CECs) were used for an in vitro cytocompatibility study, and the results showed that carbon nanocompounds in the fibers enhanced cell vitality, indicating that they might be used as an environment that is conducive to endothelial cell growth [91]. Furthermore, in order to explore the possibility of using SWNT-CS/PVA nanofibers for brain tissue engineering, Shokrgozar et al. fabricated the blended electrospun nanofibers and confirmed increased proliferation rate of both human brain-derived cells and U373 cell lines [92]. The development of NSCs can be enhanced by electrical stimulation; however, this is dependent on a stable nanopatterned electroconductive substrate. Garrudo et al. aimed to optimize the electroconductivity of polyaniline–PCL electrospun nanofibers by using the pseudo-doping effect. There were favorable impacts on doublecortin (DCX) and microtubule-associated protein 2 (MAP2) gene expression and cell alignment. The newly developed optimized platform has potential uses in the development of in vitro drug screening platforms, deep brain electrode interfaces, and transplanting fully developed and functioning neurons [93].

Every component of a composite nanofiber can be functionalized in blend systems to fulfill a particular role. For instance, one component may provide structural support, and another may contain bioactive chemicals, thereby granting the material controlled release capability. Multiple capabilities may be included into a single nanofiber system thanks to this functionalization. They have been widely studied for a drug delivery system for brain tumors. Ramachandran et al. showed a novel approach that uses a blend of PLGA–PLA–PCL polymers to target glioblastoma. This approach facilitated the delivery of the chemotherapeutic drug temozolomide (TMZ) directly to tumors in an orthotopic brain tumor model and demonstrated effective, long-term control on glioblastoma multiforme (GBM) [94]. Similarly, in different research, carmustine, a drug used in interstitial chemotherapy for GBM, was tested in coaxially electrospun fiber membranes made of the sheath polymer PCL and the core polymer poly(1,3-bis-(p-carboxyphenoxy propane)-co-(sebacic anhydride) (PCPP-SA). The use of multilayered membranes composed of core–sheath fibers can result in carmustine long-term release [95]. Also, synthetic and natural polymer composites, such as PCL and gelatin, have been successfully developed as an antitumor SN-38 loaded drug delivery platform for brain tumors and fibers, showing good biodegradation and antitumor function [96].

### 4.2. Particle-Decorated Fibers

In tissue engineering, the design of scaffolds with biomimetic and bioactive properties is highly desirable to improve the regeneration capacity of tissues. In particular, research on neural tissue regeneration has been focused on the development of biomaterials with particular topography to mimic the ECM [97,98]. Additionally, bioactive factors, such as proteins and growth factors, should be considered to promote the formation of the new tissue [99]. In this regard, the use of EFDTs can offer the possibility of combining both electrospinning and electrospraying to generate hybrid scaffolds with a controllable topography and a delivery system of bioactive molecules by adapting some parameters of process (Figure 6) [100,101]. This technological approach, inspired by additive manufacturing, allows functionalizing a fibrous network by the deposition of nanoparticles that can be optimized independently upon the surrounding substrate [45]. The large versatility of both the processes—electrospinning and electrospraying—can allow switching among different modes to integrate nanoparticles into the fibrous network (i.e., sequential or simultaneous [101]), providing different solutions to match the release mechanisms to the specific applicative demands.

More recently, a fiber scaffold decorated with collagen nanoparticles with a density gradient was fabricated by electrospraying of collagen onto aligned fibers of PCL [102]. In detail, collagen nanoparticles with a density gradient were collected onto radially aligned fibers via electrospraying by the use of a size-tunable aperture working as a mask between the needle and the grounded collector [71]. Once the hole was gradually opened, particles tended to be distributed onto more extended portions of the surrounding substrates, thus altering the particle density in time, from the center to the periphery. It was demonstrated that this peculiar configuration can promote cell migration, due to the synergic contribution of radially aligned nanofibers (e.g., topographic signal) and nanoparticle density gradient (e.g., haptotactic cue) [71].

Overall, synthetic polymers, including PCL, PLA, and PVA, are generally used for the fabrication of micro- or nanofibers due to their high mechanical stability and recognized biocompatibility, while natural polymers—being the major constituent of the ECM—are usually preferred for the fabrication of nanoparticles with relevant advantages respect to other approaches suitable to support cell–material interaction [103,104]. For instance, the combination of synthetic and natural polymers into blends can reduce the ultimate mechanical properties of the fibers [46]. In particular, wool–keratin dispersed in PCL solution tend to form an inhomogeneous mixture as a function of relative concentration (higher than 20%), thus generating heterogeneous fibers [105]. Conversely, keratose—keratin from hair oxidation—nanoparticles deposited via co-electrospraying onto PVA nanofibers do not compromise the mechanical properties of fibers, also improving adhesion and proliferation of neural cells when compared with PVA fibers [106]. From this perspective, nanoparticle-coated nanofibers may represent a potential application for neural tissue engineering. Moreover, the versatility of electrospraying allows the formation of core–shell nanoparticles able to protect and release bioactive molecules properly loaded under the fiber shell. For instance, neurotrophic factors have been loaded in poly(d,l-lactide-co-glycolide) core–shell nanospheres, then electrosprayed onto PCL aligned fibers, to obtain an integrated platform with potential use for guiding neural tissue growth and regeneration by combining both physical guidance and molecular delivery [107].

Modulation of cell migration and neurite extension play important roles in CNS cell repair [108,109]. Hence, to regulate cell migration, a class of uniaxially aligned nanofibers with nerve growth factors (NGFs) loaded into PCL microparticles was designed [110]. In this case, the long-term and sustainable release of NGF from microparticles and aligned topography were able to guide the growth of axons and migration of neural stem cells. Moreover, the modification of surface roughness by changing the deposition time of loaded microparticles influenced the axon outgrowth of PC12 and SH-SY5Y cells and the alignment of Schwann cells. To evaluate the influence of surface roughness on neurite extension, aligned fibers decorated with a moderate density of electrosprayed fatty acid microparticles provided an optimal surface roughness to promote the neurite extension of PC12 and dorsal root ganglial (DRG) cells [111].

By combining electrospinning and electrospraying, hybrid scaffolds composed of highly aligned fibers of PCL coated with electrosprayed collagen and polypyrrole nanoparticles were fabricated. The aligned topography induced the orientation and neurite/axon extension of PC12 cells along the fibers, whereas the conductive microenvironment enhanced the outgrowth of neurite/axons. The combination of both topography and conductivity enhanced neurogenesis due to the increase in voltage-gated calcium channel protein (VGCC) expression, which activated intracellular signaling [112].

### 4.3. Particle-Loaded Fibers

Nanoparticle (NP) loading within electrospun micro/nanofibers allow the obtainment of functional hybrid fibers with empowered features provided by both NPs (guest) and polymer mats (host). This strategy results in a synergistic combination between the functional features of NPs (e.g., small size, high surface:volume ratio and chemical reactivity), along with flexibility and porosity of the polymer mats. Viscosity, spinnability and resultant composite nanofiber morphology are among the main parameters affected by NP loading and distribution within the polymer solution. Therefore, an NP clustering effect needs to be avoided by means of physical methods (e.g., stirring, sonication), chemical modification (e.g., surfactants) to improve colloidality of NPs, or dissolution of NPs and polymers in different solvents. Several NPs have been successfully embedded within micro/nanofiber mats, including metal [113,114], metal oxide [114], and carbon-derived [115] and polymer-based NPs [116]. NPs can be incorporated within electrospun micro/nanofibers by three main routes. The most used is direct blending electrospinning, where the NPs are directly added to the polymer solution prior to electrospinning [116,117].

However, the distribution of NPs within the polymer solution may affect the spinnability and the resultant hybrid composite fiber morphology, leading to fewer mechanical properties and loss of functional sites in composite fiber mats. Another way to encapsulate nanoparticles is through coaxial electrospinning, creating core–shell like fibers. This approach allows the uniform embedding of pharmaceutical agents and/or NPs into the core materials, preserving them from the surrounding environment by a shell layer and leading to a sustained and kinetically controlled release over a long time. The third option involved in situ NPs growing directly within nanofibers, which can be used as substrates. This approach is typical of inorganic NP-loaded electrospun composite fibers. The process involves the dissolution of metal salt (precursors) into the polymer solution, followed by composite nanofibers mat by electrospinning and NP formation by thermal (e.g., sintering, calcination) or chemical (e.g., redox agents) means or a combination of them. The physicochemical conditions of the post-treatments, like calcination temperature and concentration of redox agents, may influence both the crystalline phase of NPs and surface morphology of the nanofibers. The latest advancements in particle-loaded electrospun composite nanofibers intended for neuronal tissue engineering and/or DDSs for brain disease treatment have been focused on improving conductivity features and functional healing. Carbon-derived NPs have been widely used as conductive fillers within electrospun composite fibers in neural engineering applications. Steel et al. described a nanofibrous conductive composite based on an ultralow concentration of carboxylated multiwalled carbon nanotubes (MWCNTs) electrospun within methacrylated hyaluronic acid (MeHA) nanofibers (MeHA/MWCNTs) [118]. The incorporation of an ultralow concentration of MWCNTs (0.01% *v*/*v*) resulted in a better dispersion within HA, a lower impedance and a higher charge capacity of HA fibers. HA-MWCNT substrates proved to firmly sustain neuron growth, as evidenced by neurite length and neuron number, for 72 h, with only 1 h of an applied 20 Hz biphasic alternating current (AC) waveform 24 h post-seeding, owing to the activation of voltage-sensitive ion channels. Additionally, da Silva, D.M. and co-workers recently explored the inclusion of poly-dopamine functionalized reduced graphene oxide (PDA@rGO) within adipose tissue-derived extracellular matrix (adECM) assisted by a lactide–caprolactone copolymer and processed into a bidimensional (2D) nanofiber platform by electrospinning intended for guiding NSC fate [119]. adECM/PLA-PDA@rGO was electrospun into a dense arrangement in a uniform and bead-free 2D nanofibrous platform, with PDA@rGO sheets homogeneously integrated within the membrane, as confirmed by SEM (Figure 7a). PDA@rGO incorporation resulted in a reduction in the average fiber diameter from 564 ± 238 nm (undoped) to 377 ± 158 nm (PDA@rGO doped), probably due to an improved conductivity of the electrospinnable solution. In fact, an increase in the electrical conductivity from 5.5·10^−6^ S·cm^−1^ (undoped) to 2.3·10^−5^ S·cm^−1^ was recorded by adding PDA@rGO. The metabolic activity/proliferation of NSCs was highly dependent on PDA@rGO presence on bidimensional platforms. NSCs exhibited the typical clustered round-shaped morphology (undifferentiated cells) after 7 days (Figure 7b) and a further growing reaching confluence after 14 days, as confirmed by phalloidin/DAPI staining (Figure 7c), whereas in the absence of exogenous diffusible signaling differentiation-inductive factors (e.g., retinoic acid and brain-derived neurotrophic factor), the presence of PDA@rGO boosts spontaneous NSC differentiation toward the neuronal lineage in 2D adECM–PLA platforms.

Other attempts have exploited hybrid electrospun composite nanofibers as delivery systems for brain cancer treatment. In this regard, Unal, S. and colleagues developed PCL–gelatin nanofibrous encapsulating bacterial cellulose nanocrystal (BCNC) as a platform for mimicking the GBM tumor ECM [120]. BCNC loading increased the fiber diameters within the nanofibrous matrix and changed the fiber morphology towards the beaded formation. PCL–gelatin–BCNC nanofibers promoted axon growth and elongation of glioblastoma cells (U251 MG), making it a potential 3D platform to support cell proliferation and adhesion. Additionally, Rasti Boroojeni et al. looked into the synergistic effect of scaffold conductivity and sustained release of thyroid hormone triiodothyronine (T3) on selective NSC differentiation toward oligodendrocyte-like cells [121]. T3-loaded CS NPs were synthesized by ionic gelation and introduced within the PCL solution, while a polyaniline–graphene (PAG) nanocomposite was prepared and incorporated into gelatin nanofibers. The two solutions were co-electrospun using two different needles on opposite sides of the electrospinning device, resulting in a composite PCL–T3@CS–gelatin–PAG nanofibers. The electrospun composite nanofibers loaded with 2% PAG and 2% T3@CS NPs exhibited the best biocompatible cellular support and proliferation with good electrical conductivity of 10.8·10^−5^ S·cm^−1^. Furthermore, PCL–T3@CS–gelatin–PAG nanofibers efficiently induced in vitro differentiation of bone marrow-derived NSCs (BM-NSCs) into oligodendrocyte-like cells, as confirmed by high level of oligodendrocyte marker expression, like O4, oligodendrocyte transcription factor (Olig2), platelet-derived growth factor receptor-alpha (PDGFR-α), O1, myelin/oligodendrocyte glycoprotein (MOG) and myelin basic protein (MBP), which might be related to sustained release of T3 over time. Molina-Peña, R. and co-workers [122] proposed a strategy for trapping GBM cells by means of a 40 μm-thick CS electrospun membrane encapsulating SF-1α-loaded PLGA-based NPs (CS/SF1-α@PLGA), showing an average fiber diameter of 261 ± 45 nm. SF1-α@PLGA NPs were clearly visible along nanofibers length as “bulges” on SEM analysis. Although the membranes lost over 10% of their initial mass after 5 weeks, they also provided a sustained release of SDF-1α. Furthermore, high cytocompatibility was confirmed in vitro by using several cell lines in particular rat primary astrocytes, showing excellent anchoring sites to support the adhesion of human GBM cells by extension of their pseudopodia. Recently, Jiang et al. exploited the possibility of developing curcumin (Curc)-loaded zeolite Y NPs (nZY) integrated into hybrid PCL–gelatin (PG) electrospun nanofibers (Curc@nZY-PG NFs) as an implantable DDS for potential post-surgical GBM treatment [123]. Curc@nZY-PG NFs exhibited slightly rough and randomly oriented fibers with a mean diameter of 640 nm. PCL, gelatin and Cur@nZY NPs strongly interacted among them by means of hydrogen bonds, which resulted in a rigid intermolecular interactive network formation. This translates to a more stable nanofibrous mat exhibiting a controlled biodegradability (63% of mass loss within three months), a steadier drug release profile over a relatively long time (47% of Curc release within 14 days) and higher tensile strength (2.75 ± 0.1 MPa) and Young’s modulus (151.3 ± 5.5 MPa). Curc@nZY-PG NFs exhibited superior cytotoxicity, anti-cell-migratory activity, and proapoptotic effect in the first 72 h against U87-MG cells compared to normal human astrocytes (NHAs). A Curc@nZY-PG NFs proapoptotic effect was also confirmed by nuclear fragmentation and upregulation of Bax, CASP-3, and CASP-9 expression levels along with a downregulation of hTERT and antiapoptotic BCL-2 expression levels in U87-MG cells. Concurrently, Yang et al. proposed near-infrared (NIR)-II-triggered composite electrospun nanofibers based on CS embedding copper-selenide (CuSe) NPs to simultaneously reach rapid intracranial hemostasis, killing superbug and residual cancer cells associated with GBM post-surgery [124]. This approach represents a prompt strategy to shorten craniotomy time, significantly reducing tumor recurrence and accelerating incision repair, through a minimally invasive surgery. Bazzazzadeh and co-workers exploited the incorporation of magnetic MIL-53 nanometal organic framework particles (NMOFs) into poly(acrylic acid) (PAA)-grafted CS–PU (PAA-g-CS/PU) core–shell nanofibers for controlled release of TMZ and paclitaxel (PTX) against U-87 MG cells in a dual chemo/hyperthermia therapy [125]. NMOF-CS-g-PAA-PTX-TMZ/PU core–shell nanofibers with an average fiber diameter of 250–300 nm were produced by coaxial electrospinning. NMOF loading increased the Brunauer–Emmett–Teller (BET) surface area of core–shell fibers from 123 to 178 m^2^·g^−1^. The nanofibers exhibited an encapsulation efficiency of TMZ and PTX higher than 80%, which were released with a maximum release rate under pH of 5.5 at 43 °C and minimum release rate under pH of 7.4 at 37 °C. The co-delivery of TMZ and PTX from the nanofibers resulted in U-87 MG cell necrosis of 56.3% (−AMF) and 68.9% (+AMF), respectively, compared to the same nanofibers without TMZ/PTX: 3.2% (CS-g-PAA/PU) and 15.2% (NMOF-CS-g-PAA/PU). In addition, the flow cytometry results indicated that 31.3% and 49.6% of apoptotic cell death occurred for U-87 MG glioblastoma cells treated with NMOF-CS-g-PAA–TMZ-PTX/PU in the absence and presence of AMF, respectively.

### 4.4. Neat and Nanocomposite Fibrogels

Over the last few years, biomaterials and hydrogels, either natural or synthetic, have been widely investigated for potential use in the brain, not only as growth factors, cell or therapeutic drug delivery carriers, but also in the development of injectable formulations and scaffolds to support and guide the growth of endogenous or exogenous cells after implantation into damaged areas of brain tissue [126,127,128,129,130,131,132]. Even though hydrogels can change the degree of swelling and/or cross-linking in their structure, they are often reported to have poor mechanical properties, with high degradation rates and with a low modulus matching the brain, which may limit their final application. However, for brain applications, they may provide a biocompatible 3D matrix with interconnected pores that fairly mimics the native ECM, with highly controlled properties like biodegradation. As a delivery platform, they may offer chemical and biological smart responsiveness to external stimuli like pH or temperature [133,134,135]. It is important to remember that the hydrogel composition can restrict the ability of cells to adhere and proliferate. Nevertheless, this can be avoided by covalently attaching peptide sequences to the polymer chains of hydrogel, which may affect the activity of embedded cells [136]. However, with the right modification, some hydrogel limitations might be easily overcome. Indeed, hydrogels can be suitably chemically modified to improve the mechanical properties and increase their residence time [137,138,139,140]. Hydrogels can be also modified by using electrospun nanofibers. In addition to mechanically strengthening the hydrogel, electrospun nanofibers provide ECM-mimicking substrates, creating an environment that perfectly replicates the organization of collagen fibers and proteoglycans in the original ECM to enhance cell adhesion and differentiation [141,142]. Indeed, due to their ability to withstand contractile stresses, they are a preferred material for cell attachment sites. Electrospun fibers are also composed, scaled, and topographically manufactured in a tailored way. Compared to other materials, electrospun nanofibers can be easily prepared using a wide range of materials, and they are characterized by good surface chemical properties of adsorbing drugs, high drug loading efficiency, and controlled drug delivery [143,144]. Above all, to meet the specific requirements for the final application, the design of injectable formulations, 2D mats or 3D scaffolds based on electrospun nanofibers involves varying their physical/topographical cues, such as diameter, alignment, density, surface chemistry, and load composition, to endow the nanofibers with different and peculiar properties [145]. Indeed, the structure and morphology of electrospun nanofibers are important factors that support and guide cell growth. At present, a variety of hydrogels and electrospun nanofibers have been widely used in brain research. Natural and synthetic materials with high biocompatibility have been applied for the fabrication of electrospun nanofibers, including collagen, HA, CS, silk, fibronectin, fibrinogen, PLLA, poly(L, D-lactide) (PLA), PCL, PLGA, polystyrene (PS), and polypyrrole (Ppy). There are two different ways to add electrospun fibers into the hydrogels (Figure 8). The first involves building a sandwich model by laminating layers of hydrogel and nanofibers. This technique allows creating a multilayered fibrous hydrogel in a repeatable, predesigned manner, but it does not offer injectability for such approach. The morphology and mechanical properties of the structure can be fine-tuned by varying the layout, classes of fibers and hydrogels employed. To increase the mechanical integrity, hydrogels are typically directly cross-linked with electrospun fibers. Hydrogel precursor solutions can be used as soak solutions to submerge fiber mats or dropped onto previously constructed fiber meshes [146,147]. The second method of hydrogel functionalization with electrospun fibers is simpler and more common. Physical methods, such as solution blending, can be applied to produce neat hybrids and nanocomposite fibrous hydrogels with homogeneous properties by dispersion of the electrospun-fibers within a hydrogel solution. Electrospun fibers—cut into short fibers—can be homogeneously dispersed in a solution using an ultrasonic homogenizer to interpenetrate and entangle through the walls and pores of the hydrogels and became completely integrated in the matrix. Afterwards, the mixture can be successively freeze-dried to obtain porous 3D sponges or aerogels [148]. Chemical cross-linking or temperature-mediated cross-linking among short fibers can be further adopted to modulate multifunctional structure features [149]. It is essential to adjust the composition of electrospun fibers to facilitate fiber dissolution and dispersion with a well-defined architecture in injectable or noninjectable hydrogel solutions [150]. Injectable therapies are beneficial because they are less invasive, which lowers the risk of infection and shortens the recovery period. In addition, injectable hydrogels play a critical role when material needs to be quickly injected into otherwise inaccessible parts of the body, such as neural tissue. Another interesting approach is adding nanofillers, such as inorganic or organic particles, to the hydrogel matrix or decorating/loading fibers with such nanoparticles. In such fibrous nanocomposite systems, hydrogels are the perfect medium for nanofiller dispersion. By boosting bioactivity, mechanical stiffness, magneto-electric features, or enabling regulated release of drugs or growth factors, nanofillers may provide a supportive role. The next sections will independently address each approach, summarizing the latest research on brain tissue applications (Figure 8).

#### 4.4.1. Multilayers

The lamination method was used by Kaixiang Huang et al. [153]. The authors electrospun PS nanofibers onto one face of a Matrigel-coated paper as a BBB model. The addition of astrocytes (iPS-Astros) increased the barrier properties of induced pluripotent stem cells (hiPSCs)-derived endothelial cells (iPS-ECs), indicating positive regulation of astrocytes to the BBB model. Further RNA-sequencing gene analysis highlighted that the introduction of iPS-Astros regulated the gene expression of the tight junction protein family and vascular endothelial (VE)-cadherin in iPS-ECs, explaining the increased trans-endothelial electrical resistance [153]. In the field of brain tissue engineering, PLLA, CS and gelatin were used to design a triple-layered biocomposite substitute of dura mater [154]. Initially, PLLA films were produced with varying *w/v* concentrations (7%, 8.5%, and 10%). PLLA–CS solution was directly spun on the PLLA film surface at different ratios (10/90, 20/80, and 30/70, *w*/*w*) to create the intermediate layer. Afterwards, a mixture of equivalent gelatin solution (15 *w*/*v*%) and CS at different concentrations (0, 1, 2, and 3 wt%) was prepared as hydrogel precursor. Small intestine submucosa (SIS) powders (0.5%, *w*/*v*) were also added. The hydrogel precursor was poured into the template [154]. Then, the double-layered electrospun membrane was cut into strips and placed on the hydrogel precursor with the CS–PLLA layer facing down. One layer of gelatin–CS hydrogel with and without acellular SIS powders was prepared for in vitro and in vivo study, respectively. Results showed that the three-layer structure was stable after 30 days in vitro. In vivo, the structure showed good stability over 14 days. Even though the tensile strength (366 ± 2 kPa) was lower than values reported for the natural tissue, it is important to remember that the dura can only withstand a maximum of 6.66 kPa of cerebrospinal fluid pressure. The PLLA layer offered excellent leakage prevention property, due to the hydrophobic properties conferred by methyl enrichment on the surface obtained during electrospinning process. In the present study, in vitro results revealed that fibroblasts were only present on the top surface of the PLLA film after 5 days of cell culture, thus indicating a satisfactory cell barrier function. The in vivo study also revealed that after 14 days, cells expanded into the hydrogel pores and occupied the degradation area, forming new tissue with newly formed vessels. In contrast, very few cells passed through the double-electrospun layer and very little collagen was deposited at the cellular infiltration sites of PLLA film. Furthermore, no overt inflammatory responses were observed against the PLLA film, ruling out the possibility of adhesion brought on by a strong inflammatory response. Additionally, the in vivo experiments demonstrated the bioactivity of hydrogels to support tissue regeneration, most likely because of the comparatively slow rate of disintegration of high-molecular-weight PLLA films and the prolonged release of SIS powders. Additionally, SIS powders induced M2-like macrophage polarization [154]. A unique technique for building a layered three-dimensional scaffold was presented by Honkamaki et al. to improve the artificial microenvironment for neurons produced from human pluripotent stem cells (hPSCs) [155]. The invention of a revolving dual-electrode collector and the assembly of fibers onto a substrate that was horizontally positioned between the electrodes were the basis for the Honkamki method’s originality. The apparatus allowed electrospinning well-aligned PLA fibers directly on top of type 1 collagen hydrogel embedding hPSC neurons. The three parallel fiber layers arranged within a 3D hydrogel matrix comprised the 3D scaffold. After 2 weeks of cell culture, the orientation of the individual neurons close to the fiber layer was analyzed to study the influence of the fibers on the directed growth of neurons. The findings demonstrated that in the absence of further functionalization, the fiber layers enhanced directed cell proliferation and neurite extension [155]. Recently, enzymatically cross-linked gelatin hydrogels with stiffness varying from 8 to 80 kPa were coupled with a sparse monolayer of PCL electrospun fibers [151]. Using electrospinning as a method to introduce anisotropic characteristics to the hydrogels, PCL fibers were spun on top of the gel in either a random or aligned way without appreciably changing the stiffness of the hydrogel substrate (Figure 8). On the fibrous hydrogel scaffolds, a human neural progenitor cell line attached, proliferated, and differentiated [151].

Microscale integration of electrospun fibers and hydrogels via a proper integration of electrospraying and air brush-spraying to deposit nanocomposite hydrogel solutions on the same collection for electrospun fibers have been also investigated [156]. For example, tri-layered zein–PVP–graphene oxide–zein nanofiber mats have been obtained by Lee et al. [157] in sandwich-layered structures with biphasic drug release behavior. In particular, the presence of graphene oxide in PVP nanofibers results in improved mechanical performance of mats and tailored release features.

#### 4.4.2. Injectable/Scaffold Systems

Hsieh et al. investigated the incorporation of electrospun fibers in hyaluronic acid and methylcellulose blended fibrous hydrogels as a potential injectable cell delivery vehicle for repair of the CNS [158]. The authors fabricated two types of electrospun fibers: genipin cross-linked collagen and a co-polymer of poly(ε-caprolactoneco-DL-lactide) (PCL-PLA). The fibers in the collected fiber mats were broken up into smaller pieces using sonication, and then the fragments were mixed with the hydrogel (5 mg/mL). With a 30 G needle, the blended fibrous hydrogel could still be injected. In fact, the hydrogel contained a good distribution of the electrospun fibers. Cells seemed to aggregate and concentrate around the fibers when mixed with neural stem/progenitor cells. The evaluation of cell proliferation revealed that the collagen electrospun fibers had a negative impact on cell viability, as evidenced by a decrease in the number of cells after 7 days in culture. Synthetic fiber composites seemed to influence cell differentiation more than their natural equivalent [158]. A hydrogel composed of 1.5% SeaPrep agarose and 7.0% Methocel methylcellulose was produced by Rivet et al. [26]. PVA electrospun fibers mats fragments were dispersed into the hydrogel matrix [26]. The agarose–methylcellulose combination was selected because of biocompatibility, physiologically relevant thermogelation characteristics, and injectability. Moreover, the materials based on cellulose did not undergo considerable degradation during the investigation, which allowed the researchers to study and distinguish the infiltration induced by hydrogel degradation from that induced by electrospun fiber topography. To demonstrate the injectability of the hybrid scaffold and determine whether the presence of fibronectin improved cellular recognition of the injected fibers, the scaffold was implanted into the rat striatum. When injected into the rat striatum, infiltrating macrophages/microglia and resident astrocytes were able to locate the fibers and use their cues for migration into the hybrid matrix [26]. Ting-Yi Wang et al. developed a composite scaffold incorporating electrospun PLA short nanofibers embedded within a thermoresponsive xyloglucan hydrogel, which could be easily injected into the injured brain (Figure 8) [152]. Moreover, glial-derived neurotrophic factor (GDNF), a protein known to support axonal growth and cell survival, was embedded within the fibrous hydrogel or covalently bonded to it to offer regulated administration and enhance the trophic qualities of the host brain. The materials’ ability to sustain ventral midbrain (VM) dopamine progenitors, continuously release GDNF, and have an impact on cell survival and dopaminergic axon formation was validated in vitro. In Parkinsonian mice, these fibrous blended hydrogels were shown to improve VM graft survival and striatal re-innervation in without adversely affecting the host immune response. This result was further increased by GDNF administration. All these findings offered a way to greatly modify the environment of the damaged brain, promoting improved graft neuron integration and survival [152]. Bruggeman et al. used self-assembling peptide (SAP) hydrogels based on a well-known peptide epitope from laminin, a common ECM protein, particularly within the CNS: fluorenylmethyloxycarbonyl (Fmoc) capped aspartic acid–isoleucine–lysine–valine–alanine–valine (DIKVAV) [159]. The peptide generated around 10 nm-diameter nanofibers, forming a macroscale supramolecular hydrogel with easily adjustable assembly-derived stiffness to suit different types of soft tissues. This is a crucial factor in guiding the survival, integration, and differentiation of endogenous cells. PLA was electrospun into nanofibers with diameters of approximately 100–2000 nm. Short fibers with a length of approximately 20–100 μm were combined with the DIKVAV hydrogel. The resulting hybrid fibrous hydrogel conserved the macroscopic characteristics of the SAP hydrogel, particularly its injectable, shear-thinning feature, which is crucial for microinjection into the brain [159]. At the nanoscale, the hybrid fibrous hydrogel highlighted a hierarchical architecture at multiple length scales, due to the range of nanofibers found in each component of the materials. The addition of short fibers to the hydrogel had only a minor impact on the mechanical properties of hydrogel, with a slight increase in stiffness observed with increasing short-fiber concentration, suggesting that the well-dispersed short fibers were only loosely interacting with the SAP network. The stability of the peptide structures was largely preserved. Multiple growth factors, including the neurotrophic growth factor GDNF and brain derived neurotrophic factor (BDNF), were able to be temporally/spatially released in controlled fashion by the SAP hydrogels. Through the combination of these materials, the authors were able to provide a range of nanofiber diameters and hence improved structural biomimicry of the ECM [159]. Recently, bone marrow mesenchymal stem cells (BMSCs) were loaded into rigid–flexible fibrous hydrogels made of electrospun nanofibers and self-adapting, injectable hydrogel. The effects of these loaded BMSCs on ischemia insult were studied [160]. Firstly, coaxial electrospinning was performed using 10% *w/v* solutions of PCL and gelatin methacryloyl (GelMA). Subsequently, 1% dibenzaldehyde-terminated polyethylene glycol (DF-PEG4000) and 3% glycol chitosan (GC) were combined (1:3) to produce a hydrogel. The concentration of nanofibers in the gel was 5 mg/mL. Based on an in vitro examination of BMSC survivability, migration, neurite growth, angiogenic potential, and paracrine effects, it was found that BMSCs loaded into fibrous hydrogels exhibited superior therapeutic efficacy to saline-laden BMSCs. Additionally, in vivo, BMSCs improved neurological impairments, decreased microglial and astrocyte overactivation, and enhanced neuronal proliferation and vascular expansion. They also greatly reduced the amount of brain edema and the infarct volume. According to bioinformatic analysis, BMSCs embedded into fibrous hydrogels were able to raise the activity of the PI3K/AKT signaling pathway by decreasing the quantity of exosomal miR-206e3p. In summary, BMSCs loaded in innovative blended fibrous hydrogels exhibited clear neuroprotective effects, reducing ischemia injury by promoting angiogenesis and neural regeneration in the brain following ischemic stroke. These findings offered a promising strategy for using cell transplantation to treat CNS diseases in clinical settings [160].

Karimi et al. developed complex structures obtained via the incorporation of magnetic short nanofibers (M.SNFs) and olfactory ecto-mesenchymal stem cells (OE-MSCs) into alginate hydrogels [161]. Wet-electrospun gelatin and superparamagnetic iron oxide nanoparticle (SPION) nanocomposite fibers were chopped and subsequently embedded in alginate hydrogels containing OE-MSCs derived from a neural crest. The results highlighted an accelerated neural-like differentiation of OE-MSCs due to the presence of SPIONs [161].

## 5. Design of Brain-Inspired Platforms via Integrated Technologies

The use of composite nanofibers in the different forms, as already extensively discussed, currently represents a valid strategy to support the regeneration of brain tissue, as summarized in Table 1.

In order to overcome some limitations of current approaches, nanofibers and nanopatterned structures obtained via EFDTs for neural application have been explored in recent literature, due to the robust support of axonal regeneration in in vivo rodent models of spinal cord injury (SCI), as well as for the possibility of tailoring substrates’ features to properly mimic the ultrastructure of natural ECM. Strategies adopted for nanocomposite substrates intended for peripheral nerve injury (PNI) could also inspire CNS application, including brain and SCI [162].

Just as an example, a core–shell nanocomposite conduit structure composed of PCL–chitosan–gelatin–Al_2_O_3_ (shell) and thermosensitive gellan–agar–polyaniline–graphene (core) has been proposed for its self-electrical capabilities via a co-electrospinning approach together with in situ chemical oxidative polymerization for polyaniline–graphene synthesis [163]. Nerve conduits are able to provide suitable environments for axonal guidance, with proper mechanical features [164].

On the other hand, self-assembly electrospinning provided different insights in 3D spongy structures obtained from the fast solidification of the nanofibers in a self-standing object, taking into account electrostatic induction and polarization of the deposited nanofibers [165,166]. The possibility to tailor pore size and structure porosity has been also investigated via the addition of conductive additives (e.g., H_3_PO_4_) in the polymeric solution to induce repulsive forces between nanofibers during the electrospinning process [167,168]. The possibility to control cell fate via a combination of cell-embedded hydrogels and short nanofibers represents a challenging method to develop valuable extracellular matrix analogues.

Patterned nanostructures can also be obtained via the combination of electrospinning and hydrogel lithography adopting PCL nanofiber sheet and PEG hydrogel [169]. PEG diacrylate was dropped onto a PCL electrospun nanofiber sheet followed by UV exposure, adopting a mask. High-porosity nanofibrous aerogels, characterized by low density and high surface area, have been fabricated through supercritical CO_2_ drying of electrospun hydrogel counterparts from polyvinyl alcohol and multiwalled carbon nanotubes, highlighting a shape-memory effect under thermal stimulus and improved mechanical properties [170].

Eom et al. developed an innovative hydrogel-assisted electrospinning process, named GelES, in which the metal collector is substituted by 3D hydrogel structure, the aim being to obtain 3D complex and tailored structures. Hydrogel features (e.g., in terms of biocompatibility and thermally reversible sol–gel transition) and nanofiber macrostructure behavior (e.g., mechanical and permeability features) make the GelES a valuable system for permeable tubular tissue structures optimization as well as for obtaining in vitro drug/cell delivery complex structures and platforms [171]. In this scenario, fused deposition modeling (FDM)-based 3D printing has allowed enhancement of the degree of freedom in hydrogel collector configurations, from bellow-shaped cylindrical structures to Y-branched or multi-bifurcated configurations, and as well as miniaturized human brain- and alveoli-like solid structures, adopting an electroconductive hydrogel connector as a further improvement of final structure features. Just as an example, the sol–gel transition of gelatin adopted as hydrogel collector could be exploited for their successive removal, thus obtaining a hollow 3D nanofiber macrostructure. Furthermore, hydrogels facilitate biomolecules and cell encapsulation, thus resulting in multiscale complex structures with slow biomolecule release (e.g., FITC–dextran) and improved biological features [171].

In vitro cell and tissue platforms can also be modeled adopting microchip-based approaches, in which ECM analogous structures and flow conditions are combined, thus providing cell niche or novel “cells on a chip” modules [172]. The direct deposition of electrospun fibers into fully sealed fluidic channel has been pursued and studied under continuous flowing conditions. A fibrous layer on the inner wall of a 3D-printed fluidic device has been obtained, thus providing interesting features on the improvement of culture conditions in microfluidic devices. Tang et al. integrated a hydrogel “perfusion” system and electrospinning to develop a “concrete” composite support for nerve repair. More specifically, SCI immune microenvironment reprogramming could be obtained through macrophage integrin receptor polarization in anti-inflammatory subtypes. The tailored release of such cytokines as cell-derived factor 1α and brain-derived neurotrophic factor are responsible for the increased recruitment and neuronal differentiation of NSCs, also promoting local blood vessel germination and nerve function recovery in a rat SCI model [173].

Recent progress in CAD-based 3D printing technology has allowed the fabrication of cellular, acellular, and hybrid scaffolds for multifunctional hybrid structures with improved features [174,175]. Electrospinning and 3D printing techniques could be combined in a different manner, i.e., adopting a kind of lamination strategy, in which nanofibers could be electrospun on 3D scaffolds and vice versa, or adopting a 3D-printed scaffold as collector [176]. Bioinks can also be obtained combining short electrospun nanofibers and hydrogel [177] or suspended in hydrogel precursors before gelation [178]. The synergistic combination of 3D-printed structure and electrospun nanofibers could represent the key to modulate multiscale structure features, thus obtaining more and more accurate ECM analogues.

## 6. Conclusions

Recent progress in nanoscience and nanotechnologies are opening interesting routes for designing materials tailored for innovative applications in the areas of disease diagnosis and therapy. In this context, EFDTs have been strongly emerging in the last decade, being a set of processing technologies to fabricate composite materials able to reproduce the chemical and physical complexity of the brain tissue, including the main features of the ECM, tightly linked to the neurons’ network functionalities. In particular, electrospinning, universally recognized as a gold-standard technique to fabricate micron/submicron fibers due to its high versatility, user-friendless, and large productivity at low cost, enables us to develop customizable membranes with unique advantages in terms of surface/volume ratio, interconnected porosity, full permeability and molecular transport. The chance to customize the experimental setups to design composite and/or hybrid materials by the combination/integration of organic/inorganic phases or particles eventually loaded with drug or bioactive agent delivering capacity and a large variety of operative fabrication methods—i.e., time and space-controlled deposition/coating, blending, inter-weaving, chemical and/or physical binding, layering—make EFDTs a promising tool for brain applications. However, some limitations still concern the entrapment efficiency, mainly related to the use of inorganic particles and the control of chemical degradation phenomena—able to significantly influence the fiber integrity and drug release mechanisms via nanoparticles. From this perspective, in vitro response of cells can be significantly influenced by different contributions, including the negative interaction of cells and/or complex macromolecules (i.e., proteins, growth factors) to the high voltages applied or the cytotoxic response of cells to the undesired release of organic solvents and/or inorganic phases. All these variables can also drastically affect in vitro release profiles of drugs/molecular signals, also limiting the capability to precisely respond to the physiological drug distribution conditions related to the rapid body fluid circulation.

In the case of the brain tissue, the lack of animal experiments makes particularly relevant a full understanding and the accurate study of the biological mechanisms occurring in vivo, thus reducing the gap between the experimental evidence and the required outputs for clinical trials. From this perspective, EFDTs can be successfully adapted for the implementation of innovative approaches based on the use of electrospun fibers in combination with bioactive molecules or the design of integrated processing approaches to validate in vitro models for brain niche and/or compartment models. For instance, the use of composite electrospun membranes was recently proposed for the development of a 2D static BBB in vitro model for the screening of drug-based therapies [23]. In vitro BBB models could help perform high-throughput drug screening with lower costs in the place of more expensive in vivo models currently used for clinical translation [179]. This approach exploits the use of nanofibrous biomimetic membranes with unique barrier functions suitable for actively supporting the biological functions of endothelial cells forming the walls of CNS blood vessels, promoting the transport of therapeutic molecules used for the treatment of cerebral disorders, and minimizing the tendency to accumulate them in the brain parenchyma. In order to improve the passage of drugs across the BBB, nanofibers could be decorated with variously functionalized particles in order to fabricate innovative patches for intranasal administration via mucosa—i.e., nose to brain—overcoming the barrier of administration from the parenteral route. Several studies have recently demonstrated that the use of nanocarriers based on polysaccharides (i.e., cellulose, CS, HA) can actively support the administration of prospective drugs with poor BBB permeability [180]. Indeed, they would allow for reaching the brain, regardless of the physicochemical characteristics of the molecule delivered, due to the peculiar properties of the matrix. In particular, CS NPs, characterized by adjuvant and immunostimulatory activity related to their innate immune responses, present positive charges of amino groups and highly bio-adhesive properties, suitable for promoting transit through the BBB, with effects on cell interactions (i.e., internalization) and drug pharmacokinetic mechanisms [181]. Multitherapeutic approaches could be also developed by adopting nanomaterials with the ability to respond to near-infrared (NIR) light stimuli. Nakielski et al. developed biomimetic nanostructures inspired by the mesoglea structure of jellyfish bells with photothermal responsiveness to NIR light for controlled drug release or photothermal (PTT) therapy [182]. The rapid expulsion of water could be obtained due to the plasmonic hydrogel–light interaction at 42 °C through electrospun membranes onto the hydrogel core.

At this stage, it is still crucial to convey technological skills and new knowledge to promote innovation. For instance, the integration of EDFTs with other processing technologies (e.g., bioprinting, microfluidic) offers the unique opportunity to better mimic composition and structural complexity of the CNS microenvironment. The synergic coupling of EFDTs with post-processing treatments (i.e., cold atmospheric plasma [183,184]) will enable the imparting of specific properties (i.e., light emitting, conductivity, optical) to fiber surfaces for the design of smart platforms (e.g., biosensors) suitable for innovative diagnostic/theranostic therapies. Additionally, the implementation of advanced tools (e.g., experimental or modeling) could be efficaciously used in vitro/in vivo to investigate specific biological mechanisms of brain associated with diffusion/molecular transport, i.e., water, aquaporin [185], or biomechanical phenomena, i.e., mechano-transduction [186]. However, the route towards the definition of medical devices is still long, and an accurate cost–benefit evaluation of EFDT processes in terms of technology transfer and process scaling up is needed prior to the production/market distribution of post-electrospinning products. For this purpose, the pursuit of a multidisciplinary approach becomes essential to reach the development of innovative therapeutic solutions that might be eminently satisfactory for translational clinical applications.

## Figures and Tables

**Figure 1 pharmaceutics-16-00134-f001:**
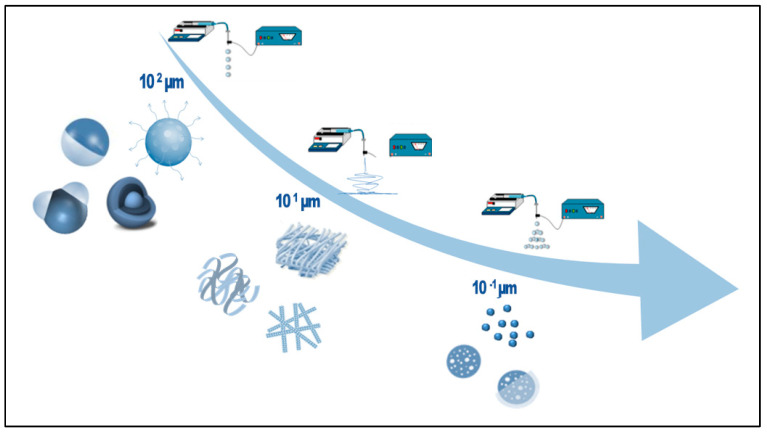
Schematic representation of multiple approaches for scaling devices fabricated via EFDTs from micro- to sub-micrometric size.

**Figure 2 pharmaceutics-16-00134-f002:**
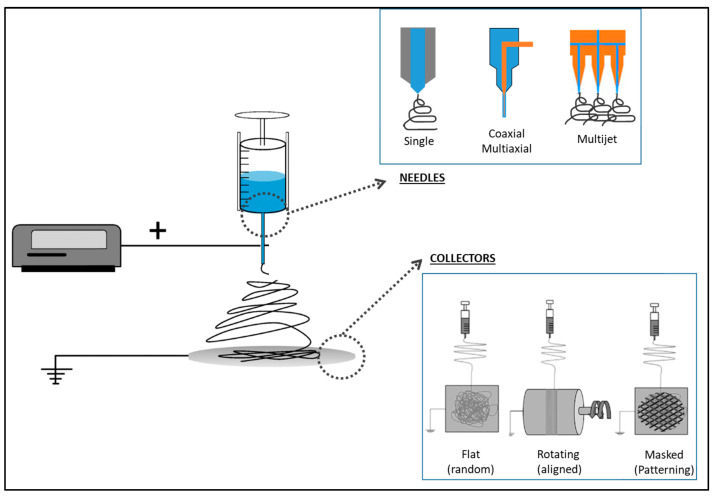
Basic scheme of the electrospinning process and setup implementation by tailored needles/collectors.

**Figure 3 pharmaceutics-16-00134-f003:**
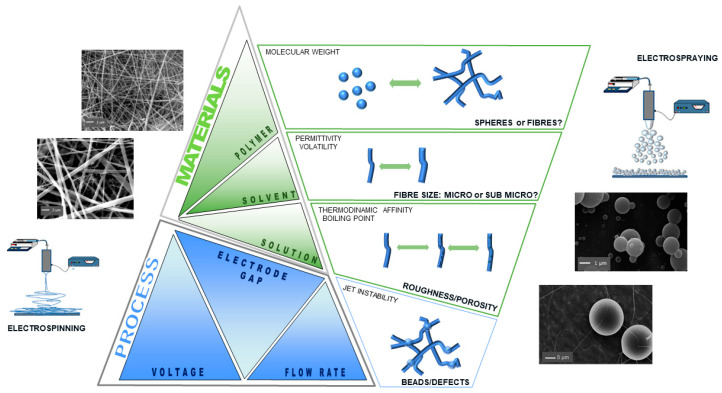
Schematic map of materials and main process parameters used to control the morphological properties of products via electrospinning and electrospraying techniques. Pictures were adapted with permission from [45,46].

**Figure 4 pharmaceutics-16-00134-f004:**
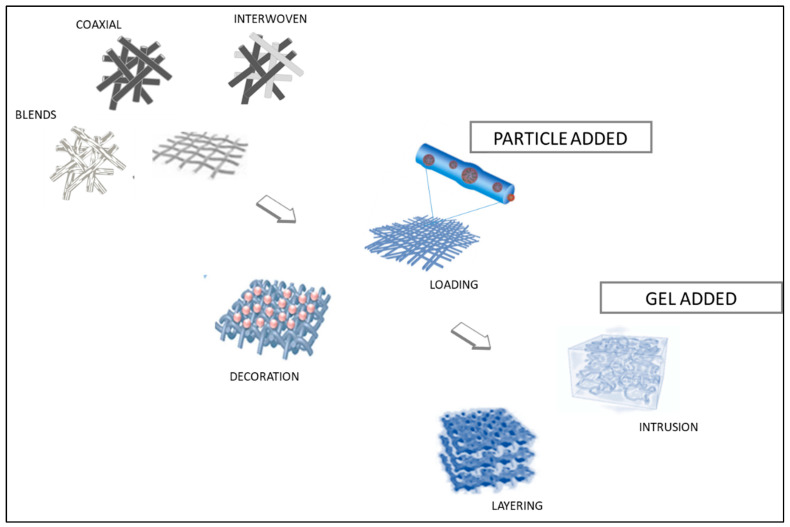
Different routes to design multicomponent and/or composite fibrous platforms via EFDTs.

**Figure 5 pharmaceutics-16-00134-f005:**
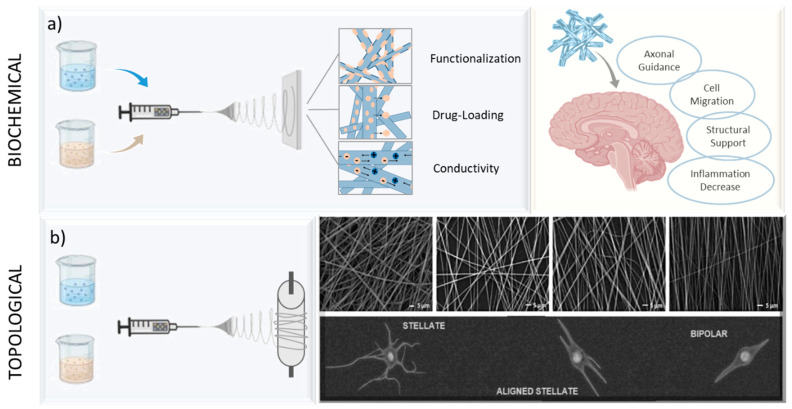
Different technological setups to impart (**a**) biochemical and/or (**b**) topographical cues in blended fibers to support brain regeneration. Pictures adapted with permission from [87].

**Figure 6 pharmaceutics-16-00134-f006:**
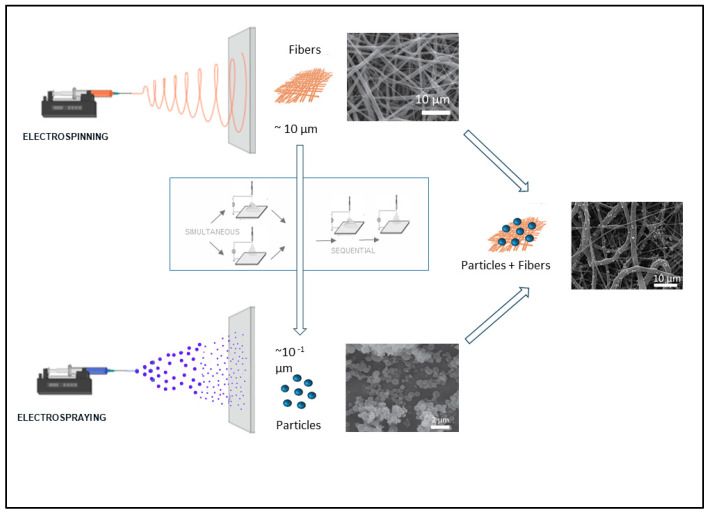
Co-electrospinning to fabricate nanoparticle decorated fibers. Particles can be collected contextually by the fibers by varying the process conditions (e.g., solution viscosity, deposition rate) to create controlled spatial gradients of drugs/molecules as a function of the particle density entrapped into the fiber network. Portions of the pictures adapted with permission from [101].

**Figure 7 pharmaceutics-16-00134-f007:**
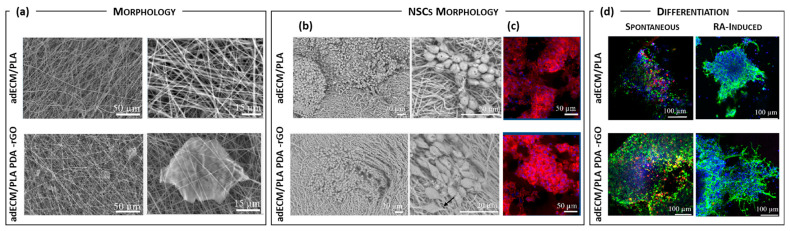
(**a**) Morphological characterization of 2D adECM–PLA and 2D adECM–PLA PDA–rGO nanofibrous platforms by SEM analysis. Morphological analysis of cytoskeleton and nuclei in NSCs grown on 2D adECM–PLA and 2D adECM–PLA PDA-rGO nanofibrous platforms by (**b**) SEM analysis (7 days post-seeding) and (**c**) immunocytochemistry (14 days post-seeding) by using phalloidin-red (actin filaments) and DAPI (nuclei). (**d**) NSC spontaneous and RA-induced differentiation on 2D adECM–PLA and 2D adECM–PLA PDA–rGO nanofibrous platforms by means of immunocytochemistry using Tuj1 (green) as neuronal marker, GFAP (blue) as astrocyte marker and DAPI (blue) for nuclei. Pictures were adapted with permission from [119].

**Figure 8 pharmaceutics-16-00134-f008:**
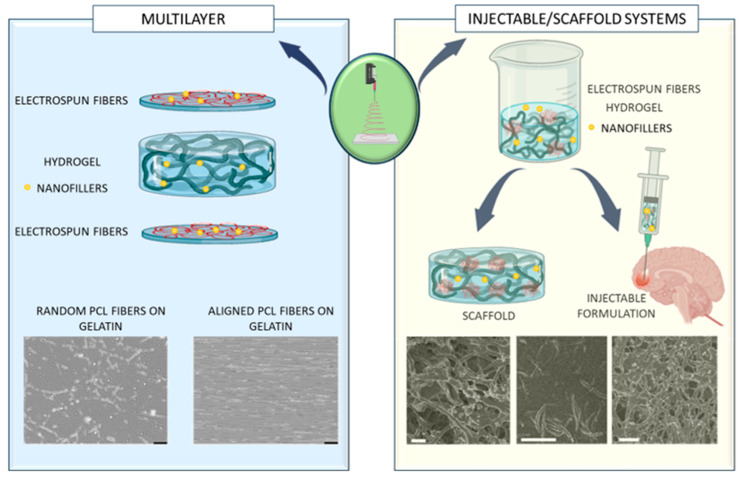
Different ways of adding electrospun nanofibers into neat and nanocomposite hydrogels to produce fibrous hydrogels: multilayer approach. (**Left**) Example of multilayer fibrous hydrogel: optical micrographs of random and aligned electrospun polycaprolactone (PCL) fibers on gelatin hydrogels. Scale bars: 100 μm. Adapted from Mungenast et al. [151]. (**Right**) Example of injectable/scaffold fibrous hydrogel. SEM images of (**Left**) xyloglucan gel with poly-D-lysine, (**Middle**) poly(L-lactide) (PLLA) short fibers, and (**Right**) the fibrous hydrogel, including PLLA short fibers within the xyloglucan gel. Scale bars: 5 μm. Adapted with permission from Wang et al. [152].

**Table 1 pharmaceutics-16-00134-t001:** Summary of composite nanofibers fabricated via EFDTs.

		Materials	Advantages		Disadvantages		Ref.
** *Neat fibers* **	Blended	PLA–PELA and PLA–PEG	- Increase in biocompatibility, - biodegradability, - hydrophilicity, - porosity		- Potential toxicity of PEG - Decreasing mechanical properties		[88]
		PCL–PHPV	- Increase in biocompatibility - Enhanced neural cell adhesion - Hydrophilicity		- Mechanical properties are not optimal		[89]
		PCL–gelatin	- Increase in biocompatibility - Enhanced astrocyte cell adhesion		- Challenge to maintain stability		[90]
		PCL–CNF and PCL–SNWT	- Increase in biocompatibility - Presence of carbon nanofillers alters surface properties, potentially improving cell response.		- Potential challenges in achieving uniform fibrous scaffolds - Agglomeration tendency of SWNTs		[91]
		CS–PVA reinforced with SNWT	- Increase in biocompatibility - Porosity - Good stability		- The need for ultrasonication may add complexity to the manufacturing process		[92]
		PCL–PANI	- Increase in electroconductivity, - Biocompatibility, - Hydrophilicity - Electroconductivity shows positive effects on cell alignment and gene expression		- Hydrophilicity changes may decrease mechanical strength		[93]
		PCL–gelatin (SN-38 loaded)	- Homogeneous nanofiber membrane - Increased swelling observed with rising SN-38 content - Good biodegradation and antitumor function		- Safety concerns related to the use of acetic acid in the fabrication process		[96]
	Coaxial	PLGA–PLA–PCL (TMZ-loaded)	- Achieving fiber-by-fiber controlled drug release with different fibers releasing the drug for specific periods, enhancing precision		- Potential challenges in maintaining the integrity and consistency of the nanofiber implant		[94]
		PCL–pCPP-SA (carmustine-loaded)	- Controlled carmustine release kinetics from the core–sheath fiber membranes - Slow polymer degradation contributes to the prolonged drug release - Incorporate different hydrophobic and hydrophilic compounds, tailoring specific drugs for a more personalized treatment		- Potential concerns regarding long-term safety and biocompatibility - Optimize the core–sheath fiber design for various drugs		[95]
** *Particle added* **	*Decorated*	Collagen nanoparticles PCL nanofibers	- Synergic contribution of radially aligned nanofibers (e.g., topographic signal) and nanoparticles density gradient (e.g., haptotactic cue)		- Particles deposited only on the top (surface contribution) - No in vitro long-term effect (due to the collagen dissolution)		[71,102]
		Keratose (oxidative keratin, KOS) nanoparticles PVA nanofibers	- Increase in hydrophilicity - Improvement of mechanical properties		- In vitro stability of nanoparticles - Not suitable for molecular release		[106]
		Poly(d, l-lactide-co-glycolide) core–shell nanospheres PCL nanofibers	- Topological signals suitable for cell guidance - Suitable for molecular delivery		- Long degradation times (not suitable for some applications)		[107]
		PCL nanofibers NGF-loaded PCL nanoparticles	- Topological signals for axons directional growth via particle gradients		- Problems of NGF stability		[110]
		Aligned PCL microfibers collagen and PPy NPs	- Topological guidance for cells - Bioactive signals to support cell adhesion - Electroconductive properties for neural signaling		- Not easy to create percolative pathways to support electroconductivity; - Difficult to encapsulate neurotrophic proteins		[112]
	*Loaded*	MeHA–MWCNTs	- Good and homogeneous dispersion within fibers - Higher mechanical features of fibers - Lower impedance and higher charge capacity of fibers. - Sustained bioactivity towards neural stem cells adhesion and migration, and boosts the differentiation towards the neuronal lineage - Remotely-controlled antitumoral activity		- NPs’ superficial defects, low surface-to-volume ratio, low reactivity, poor adhesion may affect the dispersibility of the NPs within the matrix, the viscosity of the solution and therefore the resulting mechanical properties of the composite fiber - NPs may increase the fiber diameters within the nanofibrous matrix and may change the fiber morphology towards the beaded formation		[118]
		adECM–PLA70–PDA@rGO					[119]
		CS–PEO-CuSe					[124]
		PCL–T3@CS-gelatin–PAG					[121]
		PCL–gel–BCNC	- More stable nanofibers exhibiting a controlled biodegradability, a steadier drug release profile over a relatively long time, and higher tensile strength and Young’s modulus. - Sustained and sequential release of bioactive compounds promoting axon growth or glioblastoma cells apoptosis				[120]
		CS–PLGA-PEG–SF1-α@PLGA					[122]
		Curc@nZY-PG					[123]
		NMOF-CS-g-PAA-PTX-TMZ–PU					[125]
** *Gel added* **	*Neat and Nanocomposite Multilayered*	PS nanofibers–Matrigel-coated paper	Morphology/mechanics can be fine-tuned; Hydrogels can be directly crosslinked with electrospun-fibers to increase mechanical properties; Not possible to embed cells	Hydrogel matrix: Tunable degree of swelling and/or cross-linking; biocompatibility; interconnected porosity; chemical and biological smart responsiveness to external stimuli like pH or temperature Nanofillers: Increase bioactivity; Increase mechanical stiffness; Confer magneto-electric features, Enable regulated release of drugs or growth factors Electrospun nanofibers: Mechanical reinforcement; ECM-mimicking environment. High drug loading efficiency, Controlled drug delivery	Not injectable; Delamination	Hydrogel matrix: Poor mechanical properties; High degradation rate Nanofillers: If not in the right amount, can decrease the mechanical properties, for discontinuities at particle/hydrogel interface Electrospun nanofibers: If not in the right amount or not correctly integrated, can cause poor cell infiltration and migration or delamination	[153]
		PLLA-CS nanofibers–PLLA film–gelatin-CS-SIS					[154]
		PLA nanofibers–type 1 collagen					[155]
		PCL nanofibers–gelatin					[156]
		Zein–PVP nanofibers–graphene oxide–zein					[157]
	*Neat and Nanocomposite Fibrogels*	Genipin cross-linked collagen nanofibers–hyaluronic acid-methylcellulose; PCL-PLA nanofibers–hyaluronic acid–methylcellulose	Increased homogeneity; Possibility to embed cells; Possibility to obtain 3D sponges or aerogels; Possibility to obtain less invasive therapies by using injectable formulations; Possibility to achieve inaccessible brain regions		Not easily possible to achieve aligned orientation		[158]
		PVA nanofibers–SeaPrep agarose–Methocel methylcellulose					[26]
		PLA nanofibers–xyloglucan hydrogel					[152]
		PLA nanofibers–SAP hydrogels (DIKVAV)					[159]
		PCL–GelMA nanofibers–DF-PEG4000- GC					[160]
		Gelatin-SPIONs nanofibers–alginate					[161]

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
