# Peer review of "Micro- and Nanostructured Fibrous Composites via Electro-Fluid Dynamics: Design and Applications for Brain"

_pharmaceutics, 2024, doi:10.3390/pharmaceutics16010134_

Round 1

Reviewer 1 Report

Comments and Suggestions for Authors

This is an interesting review on an important topic and details not only the fundamental EFD techniques but their application in treating disease and promoting regeneration in the brain, either alone or in combination. The review is well balanced and covers the topic in a suitable depth but the standard of English, while largely good, detracts from the review in places. I have addressed a number of issues, but by no means all, in my minor corrections below. While I believe the review is suitable for publication after these corrections have been completed, the “Conclusions and Future Perspectives” requires a significant re-write in my opinion. Rather than summarizing progress in the field and suggesting current limitations and how they may be overcome in the future, there are a number of instances where additional studies are detailed, including in the very last sentences of the review. This is not what I’d expect as a conclusion.

Moderate corrections:

p6, lines 293-296: It would be beneficial to explain how patterned particle deposition can be achieved, with the use of a figure if necessary.

p5, line 226-230: The more advanced electrospinning setups that are mentioned here are out of context and their capabilities should ideally be explained in comparison to straightforward electrospinning. A figure would be beneficial to demonstrate the different types of electrospinning.

There are a few mentions of the peripheral nervous system or sciatic nerve injuries. However, the topic of the review is “applications for brain” so the relevance of the results from these studies should be linked to their applicability in the CNS.

Minor corrections (mostly written English):

p2, line 50: Cell replacement and neuronal replacement are mentioned. Are these not the same thing or do the authors suggest that non-neuronal brain cell populations could be delivered?

p2, line 51: “has been achieved” or similar should be inserted after “success”.

p4, lines 166-8: “Continuous progresses in the biomaterial chemistry are in good agreement with the versatility of EFDT processing modes…”. This doesn’t make sense to me, especially the beginning of the sentence, which is also overlong.

p4-5, lines 184-189: The English of this section needs improving, particularly “This charged jet is then stretching and thinning…”.

p5, line 201: Should be “e.g.” rather than “i.e.”.

p5, line 203: “ES” is used without prior abbreviation.

p5, line 211: “in brain to damage” should be “to damage in the brain”.

p5, lines 220-221: “…is affect to behaviour of the human neural progenitors…” should be “…has an effect on the behaviour of human neural progenitors…”. The first part of the following sentence also needs attention.

p5, line 228: “enable to fabricate” should be “enable the fabrication of”.

p6, lines 247-248: “high-power supply” should be “high-voltage power supply”.

p6, line 260: “Although” at the beginning of the sentence is totally superfluous.

p6, line 286: “for the increase the cell viability” needs rephrasing.

P7, line 299: In the legend for Figure 2, I’m not convinced “particularize” is the right word. There is also no callout to Figure 2 in the text.

p9, line 373: It is unclear what is meant by a “nearly complex manufacturing process”. Additionally, the following sentence “Moreover, Smith et al. …” doesn’t scan.

p9, line 392: Delete “as the”.

p9, lines 396-397: “It is possible to introduce extra characteristics like conductivity by modifying its biological features.” This doesn’t make sense. Surely it’s the chemical properties which need to be modified to introduce conductivity?

p10, line 406: “enable to create a” should be changed to “enable the creation of a”.

p10, line 407: The sentence beginning “In one study…” requires rewriting as it is overlong and difficult to understand. The following sentence, beginning “According to the study…” suggests the same study is being described, but it is clearly a different paper that’s being referred to.

p10, line 427: “It was then placed into collagen tubes”. Presumably the scaffolds were placed in collagen tubes prior to implantation in a sciatic nerve injury? This isn’t clear from the text.

p10, line 433: “…with an emphasis on neurological prosthesis, the study investigates…” doesn’t make sense as no study has been mentioned.

p10, line 436: I assume that “PCL/CNS” should actually be “PCL/CNF” or “PCL/SWNT”?

p11, lines 438-440: The following sentence seems to be isolated and out of context with no further discussion of the outcomes of the work: “Furthermore, in order to explore the possibility of using -SWNT-CS/PVA nanofibers for brain tissue engineering, Shokrgozar et al. fabricated the blended electrospun nanofibers.”

p11, line 442:In the study, Garrudo et al. aimed…” - What study? Is this study meant to demonstrate an effect of a scaffold on NSCs, linking it to the preceding sentence? If so, this needs to be made clearer when mentioning the study’s outcomes.

p11, line 450: “…support the structure structurally” should be rephrased.

p11, lines 451-452: “…and it may enable them the controlled release capability.” This is poorly phrased and it surely repeats the preceding part of the sentence as they both refer controlled DDSs.

p11, line 454: The study has shown…” – Another instance of a study being referred to before any details have been given.

p11, lines 457-458: “…and effectively managed by carefully regulated, long-term control on glioblastoma multiforme (GBM)…” doesn’t make sense.

p11, lines 463-465: The following should be rephrased, ideally giving some details about biological effect (if available) or duration of release: “Another material based on electrospun has been utilised in the treatment of GBM. PVA nanofibers encapsulating dacarbazine were created, allowing the drug controlled release [115].”

p11, lines 465-466: “Also synthetic and natural polymer composites, such as PCL and gelatin, has been successfully applied for brain tumours [116].” – It would be useful to have some brief information on the findings of this paper.

p12, line 486: “Several synthetic polymers…” is a big understatement.

p12, line 493: “heterogenous” may be more appropriate than “inhomogeneous”.

p12, lines 494-495: I don’t see what the link is between the following and the sentence preceding it: “Therefore, electrospun PVA nanofibers have been coated with keratose (oxidative hair keratin) nanoparticles via electrospraying.”

p12, line 500: “For instance, aligned fibers of PCL…” – This sentence seems to be missing key information such as what the NPs were composed of and what the biological outcomes were.

p12, line 515: DRG should be defined.

p13, lines 539-540: “NPs can be incorporated within electrospun micro-/nanofibers by three main routes (Figure 6).” – This figure does not show different ways of incorporating NPs! Instead, it appears out of context and isn’t explained until ~1 page later. The correct figure, if there is one, should be placed here and figure 6 moved (and renumbered).

p14, lines 569-570: “Table XX” is mentioned but there is no table in the review. Either delete this or include the intended table.

p14, lines 599-600: “…developed a PCL/gelatin nanofibrous encapsulating bacterial…” should be “…developed PCL/gelatin nanofibers encapsulating bacterial…”

p14, lines 521-622: There are two instances of SF1-α instead of SDF1-α.

p15, line 632: “be means” should be “by means”.

p15, line 642: NIR is used without definition.

p15, lines 654-659: It would be useful to know how these results compared to monotherapy if this was tested.

p18, lines 759-760: There are two instances of KPa rather than kPa.

p18, line 788: The following sentence appears completely out of context and should be deleted: “These fibrous cell line that represents human neural progenitor cell line [173].”

p18, line 793: It is mentioned that alignment of NPC neurites can be seen in figure 7. However, this is not the case.

p19, line 833: Again, there is a claim that figure 7 shows cellular results when it clearly doesn’t.

p20, lines 878-887: “Core–shell nanocomposite…electrospinning process [186,187].” This whole section is confusing and adds nothing, in terms of positive outcomes in the brain, to the review.

p21, line 912: EDTs should be EFDTs.

p21, lines 913-915: “However, strong efforts have to be spent to solve several problems that are emerging in experimental studies.” – This line is repeated.

p21, lines 923-925: This doesn’t make sense to me: “…thus limiting the capability to precisely reply to the physiological drug distribution conditions, even partially compromised to the rapid body fluid circulation in vivo.”

p21, lines 923-925: What is meant by the following isn’t clear either: “Besides, in the case of the brain tissue, the lack of animal experiments makes particularly relevant a full understanding and the accurate study the biological mechanisms occurring in vivo, thus extending the gap between the experimental evidence and the required outputs for clinical trials.”

p21, line 946: “…poor crossing across the BBB…” should be “…poor BBB permeability…”.

p22, lines 990 & 995: SCI is used without definition.

Comments on the Quality of English Language

Please see above comments in main review.

Author Response

This is an interesting review on an important topic and details not only the fundamental EFD techniques but their application in treating disease and promoting regeneration in the brain, either alone or in combination. The review is well balanced and covers the topic in a suitable depth but the standard of English, while largely good, detracts from the review in places. I have addressed a number of issues, but by no means all, in my minor corrections below. While I believe the review is suitable for publication after these corrections have been completed, the “Conclusions and Future Perspectives” requires a significant re-write in my opinion. Rather than summarizing progress in the field and suggesting current limitations and how they may be overcome in the future, there are a number of instances where additional studies are detailed, including in the very last sentences of the review. This is not what I’d expect as a conclusion.

A0R0: The authors thank the reviewer for his/her comment. The conclusion section has been revised and some sentences were addressed to underline future progresses in the field, as follows: “At this stage, it is still crucial to convey technological skills and new knowledge to promote innovation. For instance, the integration of EDFTs with other processing technologies (e.g., bio-printing, microfluidic) offers the unique opportunity to better mimic composition and structural complexity of CNS microenvironment. The synergic coupling of EFDTs with post processing treatments (i.e., cold atmospheric plasma [184,185]) will enable to impart specific properties (i.e., light-emitting, conductivity, optical) to fibre sur-faces, for the design of smart platforms (e.g., biosensors) suitable for innovative diagnostic/theranostic therapies. Meanwhile, the implementation of advanced tools (e.g., experimental or modelling) could be efficaciously used to in vitro/in vivo investigate specific biological mechanisms of brain associated to diffusion/molecular transport (i.e., water, aquaporin [186] or biomechanical phenomena (i.e., mechano-transduction[187]). However, the route towards the definition of medical devices is still long and an accurate costs/benefit evaluation of EFDTs processes, in terms of technology transfer and process scale up, is needed prior to the production/market distribution of post-electrospinning products. For this purpose, the pursuit of a multidisciplinary approach becomes essential to reach, in a next future, the development of innovative therapeutic solutions, that might be really satisfactory for translational clinical applications.

Moderate corrections:

R1C1: p6, lines 293-296: It would be beneficial to explain how patterned particle deposition can be achieved, with the use of a figure if necessary.

A1R1: The authors thank the reviewer for his/her comment. according with the suggestion, the figure 5 has been modified and a more accurate description of the deposition methods was added in the paragraph 4.2, as follows:

“In this regard, the use of EFDTs can offer the possibility of combining both electrospinning and electrospraying to generate hybrid scaffolds with a controllable topography and a de-livery system of bioactive molecules by adapting some parameters of process (Figure 6) [100,101]. This technological approach, inspired by additive manufacturing, allows functionalizing a fibrous network by the deposition of nanoparticles that can be optimized in-dependently upon the surrounding substrate [45]. The large versatility of both the pro-cesses – namely electrospinning and electrospraying – can allow switching among different modes to integrate nanoparticles into the fibrous network (i.e., sequential or simultaneous [101]), providing different solutions to match the release mechanisms to the specific applicative demands.”.

“More recently, a fiber scaffold decorated with collagen nanoparticles with a density gradient was fabricated by electrospraying of collagen onto aligned fibers of PCL [102]. In detail, collagen nanoparticles with a density gradient were collected onto radially aligned fibers via electrospraying by the use of a size-tunable aperture working as a mask between the needle and the grounded collector [71]. Once the hole was gradually opened, particles tended to be distributed onto more extended portions of the surrounding substrates thus altering the particle density in time, from the centre to the periphery. It was demonstrated that this peculiar configuration can promote cell migration, due to the synergic contribu-tion of radially aligned nanofibers (e.g., topographic signal) and nanoparticles density gradient (e.g., haptotactic cue) [71]”.

R1C2: p5, line 226-230: The more advanced electrospinning setups that are mentioned here are out of context and their capabilities should ideally be explained in comparison to straightforward electrospinning. A figure would be beneficial to demonstrate the different types of electrospinning.

A1R2: The authors thank the reviewer for his/her comment. In the section 2, it has been included a new figure (now figure 2) where different advanced configurations were schematically described as a function of specific needle/collector used. The text of section 2 has been revised accordingly in different points.

R1C3: There are a few mentions of the peripheral nervous system or sciatic nerve injuries. However, the topic of the review is “applications for brain” so the relevance of the results from these studies should be linked to their applicability in the CNS.

A1R3: The authors thank the reviewer for his/her suggestion and agree with him/her. Therefore, the parts related to the PNS have been removed from the text.

Minor corrections (mostly written English):

R1C4: p2, line 50: Cell replacement and neuronal replacement are mentioned. Are these not the same thing or do the authors suggest that non-neuronal brain cell populations could be delivered?

A1R4: The authors thank the reviewer for his/her comment. Cell replacement is mentioned here as a general term. But that part has been deleted to make it clearer.

R1C5: p2, line 51: “has been achieved” or similar should be inserted after “success”.

A1R5: The authors thank the reviewer for his/her comment. It has been inserted.

R1C6: p4, lines 166-8: “Continuous progresses in the biomaterial chemistry are in good agreement with the versatility of EFDT processing modes…”. This doesn’t make sense to me, especially the beginning of the sentence, which is also overlong.

A1R6: The authors thank the reviewer for his/her comment. The sentence has been changed as follows:

“The high versatility of EFDT processing modes well fits to the use of biomaterials with a high tailored chemistry, thus providing a fine matching between structural and functional properties. This allows for the development of advanced composite devices for applications such as repair, regeneration, drug delivery, biosensing and, diagnostic/theranostic purposes.”

R1C7: p4-5, lines 184-189: The English of this section needs improving, particularly “This charged jet is then stretching and thinning…”.

A1R7: The authors thank the reviewer for his/her comment. The English of the section has been improved with minimal changes for improving the clarity while maintaining the original content.

R1C8: p5, line 201: Should be “e.g.” rather than “i.e.”.

A1R8: The authors thank the reviewer for his/her comment. The text has been checked completely and necessary corrections has been made.

R1C9: p5, line 203: “ES” is used without prior abbreviation.

A1R9: The authors thank the reviewer for his/her comment. Abbreviation has been added in the introduction section.

R1C10: p5, line 211: “in brain to damage” should be “to damage in the brain”.

A1R10: The authors thank the reviewer for his/her comment. It has been changed.

R1C11: p5, lines 220-221: “…is affect to behaviour of the human neural progenitors…” should be “…has an effect on the behaviour of human neural progenitors…”. The first part of the following sentence also needs attention.

A1R11: The authors thank the reviewer for his/her comment. It has been changed.

R1C12: p5, line 228: “enable to fabricate” should be “enable the fabrication of”.

A1R12: The authors thank the reviewer for his/her comment. It has been changed.

R1C13: p6, lines 247-248: “high-power supply” should be “high-voltage power supply”.

A1R13: The authors thank the reviewer for his/her comment. It has been changed.

R1C14: p6, line 260: “Although” at the beginning of the sentence is totally superfluous.

A1R14: The authors thank the reviewer for his/her comment. It has been deleted.

R1C15: p6, line 286: “for the increase the cell viability” needs rephrasing.

A1R15: The authors thank the reviewer for his/her comment. However, that sentence and reference has been deleted in accordance with R1C3.

R1C16: P7, line 299: In the legend for Figure 2, I’m not convinced “particularize” is the right word. There is also no call out to Figure 2 in the text.

A1R16: The authors thank the reviewer for his/her comment. In line with some changes and additions Figure2 has been turned to Figure 3. And “Particularize” has been changed with “control”. And callout to has been added into the previous paragraph and also into the section 2.2.

R1C17: p9, line 373: It is unclear what is meant by a “nearly complex manufacturing process”. Additionally, the following sentence “Moreover, Smith et al. …” doesn’t scan.

A1R17: In these references, multi-component fibre meshes obtained using different setups are mentioned. Illner et al. had been tried a complex process method using both different polymer solutions and different setups in the same study. But to make it clearer, “nearly” has been deleted from the sentence. Smith et.al used a combination of additive manufacturing methods with electrospinning to obtain electrospun fibres with multiple components. Here we tried to mention different combination of electrospinning for multicomponent fibers.

R1C18: p9, line 392: Delete “as the”.

A1R18: The authors thank the reviewer for his/her comment. It has been deleted.

R1C19: p9, lines 396-397: “It is possible to introduce extra characteristics like conductivity by modifying its biological features.” This doesn’t make sense. Surely, it’s the chemical properties which need to be modified to introduce conductivity?

A1R19: The authors thank the reviewer for his/her comment. The sentence has been changed with “It is possible to introduce extra characteristics like conductivity to affect biological activity by modifying fiber chemical features.”

R1C20: p10, line 406: “enable to create a” should be changed to “enable the creation of a”.

A1R20: The authors thank the reviewer for his/her comment. It has been changed.

R1C21: p10, line 407: The sentence beginning “In one study…” requires rewriting as it is overlong and difficult to understand. The following sentence, beginning “According to the study…” suggests the same study is being described, but it is clearly a different paper that’s being referred to.

A1C21: The authors thank the reviewer for his/her suggestion. Sentence has been divided as follows:

“One study describes the fabrication process and emphasises the possible applications of poly (lactic acid) (PLA)/poly (lactide-b-ethylene glycol-b-lactide) block copolymer (PELA) and PLA/polyethylene glycol (PEG) blends on the morphology, wettability, and mechanical properties of the material and the behaviour of neural stem cells (NSCs). The findings suggest that electrospun blend of PLA and PELA have favourable surface characteristics, potentially resembling brain structures [106].”

R1C22: p10, line 427: “It was then placed into collagen tubes”. Presumably the scaffolds were placed in collagen tubes prior to implantation in a sciatic nerve injury? This isn’t clear from the text.

A1R22: The authors thank the reviewer for his/her comment. This reference has been removed from the text to ensure that the study related to brain applications in accordance with R1C3.

R1C23: p10, line 433: “…with an emphasis on neurological prosthesis, the study investigates…” doesn’t make sense as no study has been mentioned.

A1R23: The authors thank the reviewer for his/her comment. The study has been mentioned at the end of following sentence.

R1C24: p10, line 436: I assume that “PCL/CNS” should actually be “PCL/CNF” or “PCL/SWNT”?

A1R24: The authors thank the reviewer for his/her comment. CNS has been used to generically indicate both components. In order to avoid confusion, text has been changed using “carbon nanocompounds”.

R1C25: p11, lines 438-440: The following sentence seems to be isolated and out of context with no further discussion of the outcomes of the work: “Furthermore, in order to explore the possibility of using -SWNT-CS/PVA nanofibers for brain tissue engineering, Shokrgozar et al. fabricated the blended electrospun nanofibers.”

A1R25: The authors thank the reviewer for his/her comment. The sentence has been changed as follows:

 “Furthermore, in order to explore the possibility of using SWNT-CS/PVA nanofibers for brain tissue engineering, Shokrgozar et al. fabricated the blended electrospun nanofibers and confirmed increased proliferation rate of both human brain-derived cells and U373 cell lines.”

R1C26: p11, line 442: “In the study, Garrudo et al. aimed…” - What study? Is this study meant to demonstrate an effect of a scaffold on NSCs, linking it to the preceding sentence? If so, this needs to be made clearer when mentioning the study’s outcomes.

A1R26: The authors thank the reviewer for his/her comment. It has been changed.

R1C27: p11, line 450: “…support the structure structurally” should be rephrased.

A1R27: Sentence has been changed as follows:

 “For instance, one component may provide structural support, and another may contain bioactive chemicals, thereby granting the material controlled release capability.”

R1C28: p11, lines 451-452: “…and it may enable them the controlled release capability.” This is poorly phrased and it surely repeats the preceding part of the sentence as they both refer controlled DDSs.

A1R28: Sentence has been changed as follows:

 “For instance, one component may provide structural support, and another may contain bioactive chemicals, thereby granting the material controlled release capability.”

R1C29: p11, line 454: The study has shown…” – Another instance of a study being referred to before any details have been given.

A1R29: The authors thank the reviewer for his/her comment. Referenced author name has been added.

R1C30: p11, lines 457-458: “…and effectively managed by carefully regulated, long-term control on glioblastoma multiforme (GBM)…” doesn’t make sense.

A1R30: The authors thank the reviewer for his/her comment. The sentence has been changed as follows:

 “Ramachandran et. al. showed a novel approach that uses blend of PLGA-PLA-PCL polymers to target glioblastoma. This approach facilitated the delivery of chemotherapeutic drug temozolomide (TMZ) directly to tumours in an orthotopic brain tumour model, demonstrated effective, long-term control on glioblastoma multiforme (GBM)[113].”

R1C31: p11, lines 463-465: The following should be rephrased, ideally giving some details about biological effect (if available) or duration of release: “Another material based on electrospun has been utilised in the treatment of GBM. PVA nanofibers encapsulating dacarbazine were created, allowing the drug controlled release [115].”

A1R31: The authors thank the reviewer for his/her comment. The sentence has been changed as follows:

 “Another approach has been studied for the treatment of GBM by developing PVA nanofibers containing dacarbazine (antitumor agent). These nanofibers demonstrated stability and  sustained drug release and enhanced its antitumor effect [115].”

R1C32: p11, lines 465-466: “Also synthetic and natural polymer composites, such as PCL and gelatin, has been successfully applied for brain tumours [116].” – It would be useful to have some brief information on the findings of this paper.

A1R32: The authors thank the reviewer for his/her comment. More information has been added.

R1C33: p12, line 486: “Several synthetic polymers…” is a big understatement.

The authors thank the reviewer for his/her comment. Synthetic polymers mainly used for the mentioned applications have been added in parentheses.

R1C34: p12, line 493: “heterogenous” may be more appropriate than “inhomogeneous”.

The authors thank the reviewer for his/her comment. The word has been changed as suggested.

R1C35: p12, lines 494-495: I don’t see what the link is between the following and the sentence preceding it: “Therefore, electrospun PVA nanofibers have been coated with keratose (oxidative hair keratin) nanoparticles via electrospraying.”

The authors thank the reviewer for his/her comment. The sentence has been changed as follows: Contrariwise, keratose – namely keratin form hair oxidation - nanoparticles deposited via co-electrospraying onto PVA nanofibers do not compromise the mechanical properties of fibers, also improving adhesion and proliferation of neural cells, when compared with PVA fibers [106].

R1C36: p12, line 500: “For instance, aligned fibers of PCL…” – This sentence seems to be missing key information such as what the NPs were composed of and what the biological outcomes were.

The authors thank the reviewer for his/her comment. the sentence has been changed as follows: “neurotrophic factors have been loaded in poly (d, l-lactide-co-glycolide) core-shell nano-spheres, then electrosprayed onto the PCL aligned fibers, to obtain an integrated platform with potential use for guiding neural tissue growth and regeneration by combining both physical guidance and molecular delivery [107]”

R1C37: p12, line 515: DRG should be defined.

The authors thank the reviewer for his/her comment. the acronym now is described

R1C38: p13, lines 539-540: “NPs can be incorporated within electrospun micro-/nanofibers by three main routes (Figure 6).” – This figure does not show different ways of incorporating NPs! Instead, it appears out of context and isn’t explained until ~1 page later. The correct figure, if there is one, should be placed here and figure 6 moved (and renumbered).

A1R38: The authors thank the reviewer for his/her comment. The authors removed “(Figure 6)” in the line 549 and move the Figure 6 (and has been turned to Figure7 in accordance with previous changes in the text) in the line 552.

R1C39: p14, lines 569-570: “Table XX” is mentioned but there is no table in the review. Either delete this or include the intended table.

A1R39: The authors thank the reviewer for his/her comment. “(Table XX)” was a mistake and it was removed.

R1C40: p14, lines 599-600: “…developed a PCL/gelatin nanofibrous encapsulating bacterial…” should be “…developed PCL/gelatin nanofibers encapsulating bacterial…”

A1R40: The authors thank the reviewer for his/her comment. Text has been revised accordingly.

R1C41: p14, lines 521-622: There are two instances of SF1-α instead of SDF1-α.

A1R41: The authors thank the reviewer for his/her comment.  It was corrected.

R1C42: p15, line 632: “be means” should be “by means”.

A1R42: The authors thank the reviewer for his/her comment. It was corrected.

R1C43: p15, line 642: NIR is used without definition.

A1R43: The authors thank the reviewer for his/her comment. It was corrected.

R1C44: p15, lines 654-659: It would be useful to know how these results compared to monotherapy if this was tested.

A1R44: The authors thank the reviewer for his/her comment. The authors improved this part adding the following sentence at lines 676-684:

“The co-delivery of TMZ and PTX from the nanofibers resulted in U-87 MG cell necrosis of 56.3% (- AMF) and 68.9% (+AMF), respectively, compared to the same nanofibers without TMZ/PTX: 3.2% (CS-g-PAA/PU) and 15.2% (NMOF-CS-g-PAA/PU). In addition, the flow cytometry results indicated that 31.3% and 49.6% of apoptosis cell death was occurred for U-87 MG glioblastoma cells treated with NMOF-CS-g-PAA–TMZ-PTX/PU in the absence and presence of AMF, respectively.”

R1C45: p18, lines 759-760: There are two instances of KPa rather than kPa.

A1R45: The authors thank the reviewer for his/her comment. It has been changed.

R1C46: p18, line 788: The following sentence appears completely out of context and should be deleted: “These fibrous cell line that represents human neural progenitor cell line [173].”

A1R46: The authors thank the reviewer for his/her comment. It has been deleted that effectively was out of context.

R1C47: p18, line 793: It is mentioned that alignment of NPC neurites can be seen in figure 7. However, this is not the case.

A1R47: The authors thank the reviewer for his/her comment. They agree and correctly reported the details. The SEM image reports only the morphology of the fibers, which result aligned or randomly organized on the gelatin.

R1C48: p19, line 833: Again, there is a claim that figure 7 shows cellular results when it clearly doesn’t.

A1R48: The authors thank the reviewer for his/her comment. They corrected the mistake. The citation to figure 7 was wrongly reported on page 19, line 833.

R1C49: p20, lines 878-887: “Core–shell nanocomposite…electrospinning process [186,187].” This whole section is confusing and adds nothing, in terms of positive outcomes in the brain, to the review.

A1R49: The authors thank the reviewer for his/her comment. That paragraph has been removed.

R1C50: p21, line 912: EDTs should be EFDTs.

A1R50: The authors thank the reviewer for his/her comment. It has been changed.

R1C51: p21, lines 913-915: “However, strong efforts have to be spent to solve several problems that are emerging in experimental studies.” – This line is repeated.

A1R51: The authors thank the reviewer for his/her comment. The line has been changed.

R1C52: p21, lines 923-925: This doesn’t make sense to me: “…thus limiting the capability to precisely reply to the physiological drug distribution conditions, even partially compromised to the rapid body fluid circulation in vivo.”

A1R52: The authors thank the reviewer for his/her comment. The line has been changed.

R1C53: p21, lines 923-925: What is meant by the following isn’t clear either: “Besides, in the case of the brain tissue, the lack of animal experiments makes particularly relevant a full understanding and the accurate study the biological mechanisms occurring in vivo, thus extending the gap between the experimental evidence and the required outputs for clinical trials.”

A1R53: The authors thank the reviewer for his/her comment. The lines have been changed to make it clearer.

R1C54: p21, line 946: “…poor crossing across the BBB…” should be “…poor BBB permeability…”.

A1R54: The authors thank the reviewer for his/her comment. It has been changed.

R1C55: p22, lines 990 & 995: SCI is used without definition.

A1R55: The authors thank the reviewer for his/her comment. The definition has been added.

Reviewer 2 Report

Comments and Suggestions for Authors

The article entitled “Micro and Nanostructured Fibrous Composites Via Electro-Fluid Dynamics: Design and Applications for Brain" provides a detailed overview a different approach for electrospinning electrospraying and related techniques, which offer the opportunity to engineer a composite substrate, by integrating fibers, particles, and hydrogels. A review also overviews different technological approaches for engineering fibrous and/or particle-loaded composite substrates. The present review also shows a current and future approaches to the use of composites for brain applications, ranging from therapeutic to diagnostic use.

The review is written to a particularly high standard, provides a thorough overview of the selected area, and only contains a very limited number of typos (e.g. line 66: bioprinted, line 108: micro-electromechanical systems (MEMS)). Regarding one shortcoming, I would suggest that you reconsider the addition of the announcement: given that the presented technique can be implemented in continuous operation, many questions arise regarding the processing and follow-up of the electrospinning process. I might suggest a brief discussion of this.

Comments on the Quality of English Language

It contains a very limited number of typos (e.g. line 66: bioprinted, line 108: micro-electromechanical systems (MEMS))

Author Response

Reviewer 2

R2C1: The article entitled “Micro and Nanostructured Fibrous Composites Via Electro-Fluid Dynamics: Design and Applications for Brain" provides a detailed overview a different approach for electrospinning electrospraying and related techniques, which offer the opportunity to engineer a composite substrate, by integrating fibers, particles, and hydrogels. A review also overviews different technological approaches for engineering fibrous and/or particle-loaded composite substrates. The present review also shows a current and future approaches to the use of composites for brain applications, ranging from therapeutic to diagnostic use. The review is written to a particularly high standard, provides a thorough overview of the selected area, and only contains a very limited number of typos (e.g. line 66: bioprinted, line 108: micro-electromechanical systems (MEMS)). Regarding one shortcoming, I would suggest that you reconsider the addition of the announcement: given that the presented technique can be implemented in continuous operation, many questions arise regarding the processing and follow-up of the electrospinning process. I might suggest a brief discussion of this.

A2C2: The authors thank the reviewer for the positive comments. Typos and mistakes have been corrected. As for the follow up of the electrospinning process, this is a relevant aspect for a perspective use of these techniques for clinical applications. In this view, a sentences have been included in the conclusion section, in order to emphasise this aspect, as follows: “However, the route towards the definition of medical devices is still long and an accurate costs/benefit evaluation of EFDTs processes, in terms of technology transfer and process scale up, is needed prior to the production/market distribution of post-electrospinning products. For this purpose, the pursuit of a multidisciplinary approach becomes essential to reach, in a next future, the development of innovative therapeutic solutions, that might be really satisfactory for translational clinical applications”.

Reviewer 3 Report

Comments and Suggestions for Authors

Dear authors

Despite that there is a great number of reviews on this topic, the review is interesting, relevant, well-written, and very complete. I recommend for publication after these corrections

a) All information in the introduction section is interesting but is too long, please adjust the introduction to at least one page and a half.

b) Can you expand on the disadvantages and limitations of the EFDTs in their respective sections, please? To discuss advantages and disadvantages

c) For Figure 2, please switch the clear tags for darker ones, because it is difficult to read

d) Please re-write the following text in the conclusion section, is very repetitive: 

However, strong efforts have to be spent to solve several problems that are emerging in experimental studies. However, strong efforts have still to be spent to solve several problems that are emerging in the experimental studies. Among them, there are some imitations that need to be necessarily improved in terms of entrapment efficiency, mainly related to inorganic particles - and chemical degradation - involving the fiber integrity and the control of drug release via nanoparticles.

Comments on the Quality of English Language

Minimal grammar or typing errors

Author Response

Reviewer 3

Despite that there is a great number of reviews on this topic, the review is interesting, relevant, well-written, and very complete. I recommend for publication after these corrections

R3C1: All information in the introduction section is interesting but is too long, please adjust the introduction to at least one page and a half.

A3C1: The authors thank the reviewer for his/her comment. The Introduction length has been reduced as requested, less than 2 pages.

R3C2: Can you expand on the disadvantages and limitations of the EFDTs in their respective sections, please? To discuss advantages and disadvantages

A3C2: The authors thank the reviewer for his/her comment. In the paragraph 2 (2.1 and 2.2), some comments about disadvantages and limitations of the EFDTs electrospinning technique, addressed to the use of synthetic/natural polymer solutions -  have been included as requested. Moreover, a new Table (Table 1) has been included to summarize advantages/disavantages of different EFDTs systems discussed into the manuscript. New sentences have been written as follows:

In the section 2.1:

“The use of synthetic polymers in electrospinning is advantageous due to their mechanical stability, biocompatibility, biodegradability, and non-toxicity, making them suitable for biomedical applications. However, many synthetic polymers usually processed by electrospinning often require the use of chemically aggressive organic solvents, with relevant limitations in the manufacturing of cell friendly substrates. Accordingly, main constrains associated with the use of toxic solvents in electrospinning are paving the way to the use of less or non-toxic solvents (e.g. water, ionic liquids), so limiting the use to a restricted group of green polymers with more sustainable properties (e.g. PVA, PEO) Alternatively, they are increasingly used in combination with natural polymers such as proteins of polysaccharides with native functions biological recognition [42,43]”

In section 2.2:

“Electrospraying is a one-step process, compatible with different biomaterials (e.g., natural and synthetic polymers) that can act as carriers of a wide variety of molecules and drugs (e.g., water or poor-water soluble drugs) [63, 64]. Congruently with electrospinning case, the use of organic solvents in electrospraying poses some limitations as it can potentially harm the bioactivity of biomolecules (i.e., proteins, genes, enzymes), also compromising the transmembrane interaction with cells. However, the electrospray is a process mainly driven by the solvent evaporation, so that the use of harmless or aqueous solvents impose to adopt further solutions - such as the use of additives or other compounds - to promote an efficient interactions of polymer solution with electrical forces, in the face of some effects in terms of particle size - only micrometric one - and shape - not spherical/eccentric [62].”

R3C4: For Figure 2, please switch the clear tags for darker ones, because it is difficult to read

A3C4: The authors thank the reviewer for his/her comment. Figure 2 has been revised according to the suggestions.

R3C5: Please re-write the following text in the conclusion section, is very repetitive: 

However, strong efforts have to be spent to solve several problems that are emerging in experimental studies. However, strong efforts have still to be spent to solve several problems that are emerging in the experimental studies. Among them, there are some imitations that need to be necessarily improved in terms of entrapment efficiency, mainly related to inorganic particles - and chemical degradation - involving the fiber integrity and the control of drug release via nanoparticles.

A3C5: The authors thank the reviewer for his/her comment. The sentence has been rewritten as follows: “However, some limitations still concern the entrapment efficiency, mainly related to the use of inorganic particles and the control of chemical degradation phenomena – able to significantly influence the fiber integrity and drug release mechanisms via nanoparticles”.

Reviewer 4 Report

Comments and Suggestions for Authors

Although the paper is interesting, it should be improved according to following lines:

1- Authors should provide a table regarding chemistry of composites that have been used by electrospinning process. Also two column should be added regarding advantages and disadvantages of that article and product.

2-Authors should provide a table regarding chemistry of composites that have been used by Electrospraying process. Also two column should be added regarding advantages and disadvantages of that article and product.

3- A table summarizing Particle Decorated Fibers based on specific particle type should be represented. What would be the advantage of using those particles?

4- A table summarizing Neat and Nanocomposite Fibrogels based on specific gel type should be represented. What would be the relations of using those gels for nerve cell regeneration? 

5- Has there been any surface modification methods to attract more cells? 

In recent years, plasma surface modification has attracted much attention for adhesion of cells. Authors should add more references regrading this process. Some articles to consider:

+ Thin film plasma functionalization of polyethylene terephthalate to induce bone‐like hydroxyapatite nanocrystals Plasma Processes and Polymers 11 (1), 37-43 (2014).
+ Surface modified electrospun nanofibrous scaffolds for nerve tissue engineering 2008 Nanotechnology 19 455102
+ Plasma surface modification of poly (l-lactic acid) and poly (lactic-co-glycolic acid) films for improvement of nerve cells adhesion Radiation Physics and Chemistry Volume 77, Issue 3, March 2008, Pages 280-287
+ The use of air plasma in surface modification of peripheral nerve conduits Acta Biomaterialia Volume 6, Issue 6, June 2010, Pages 2066-2076

Author Response

Although the paper is interesting, it should be improved according to following lines:

R4C1: Authors should provide a table regarding chemistry of composites that have been used by electrospinning process. Also two column should be added regarding advantages and disadvantages of that article and product.

R4C2: Authors should provide a table regarding chemistry of composites that have been used by Electrospraying process. Also two column should be added regarding advantages and disadvantages of that article and product.

R4C3:- A table summarizing Particle Decorated Fibers based on specific particle type should be represented. What would be the advantage of using those particles?

R4C4:- A table summarizing Neat and Nanocomposite Fibrogels based on specific gel type should be represented. What would be the relations of using those gels for nerve cell regeneration?

A1C1 to C4: The authors thank the reviewer for his/her comment. A unique table summarizing the advantages/disadvantages of all the composite nanofibers discussed in the manuscript (i.e., electrospun fibers, particle added, and gell added) was included, as follows:

R5C5: Has there been any surface modification methods to attract more cells?

In recent years, plasma surface modification has attracted much attention for adhesion of cells. Authors should add more references regrading this process. Some articles to consider:

+ Thin film plasma functionalization of polyethylene terephthalate to induce bone‐like hydroxyapatite nanocrystals Plasma Processes and Polymers 11 (1), 37-43 (2014).

+ Surface modified electrospun nanofibrous scaffolds for nerve tissue engineering 2008 Nanotechnology 19 455102

+ Plasma surface modification of poly (l-lactic acid) and poly (lactic-co-glycolic acid) films for improvement of nerve cells adhesion Radiation Physics and Chemistry Volume 77, Issue 3, March 2008, Pages 280-287

+ The use of air plasma in surface modification of peripheral nerve conduits Acta Biomaterialia Volume 6, Issue 6, June 2010, Pages 2066-2076

A2C5: The authors thank the reviewer for his/her comment. Plasma treatment has been used in the last two decades as useful strategy to improve cell attachment for different tissues (i.e., Bone, peripheral nerve) as well remarked by the references reported by the reviewer. More recently, a growing interest is emerging in the use of plasma treatment to graft molecular cues able to add more functionalities suitable for novel diagnostic approaches in CNS treatment. Accordingly, we have included a sentence in the conclusion section to highlight this aspect, reporting two new references published in the last three years: ” The synergic coupling of EFDTs with post processing treatments (i.e., cold atmospheric plasma [184,185]) will enable to impart specific properties (i.e., light-emitting, conductivity, optical) to fibre surfaces, for the design of smart platforms (e.g., biosensors) suitable for innovative diagnostic/theranostic therapies”.

Round 2

Reviewer 4 Report

Comments and Suggestions for Authors

It is acceptable now.